



# Origin of low-tropospheric potential vorticity in Mediterranean cyclones

Alexander Scherrmann[1], Heini Wernli[1], and Emmanouil Flaounas[2]

[1]ETH Zurich, Institute for Atmospheric and Climate Science, Zurich, Switzerland
[2]Hellenic Centre for Marine Research, (HCMR), Anavyssos, Greece

**Correspondence:** Alexander Scherrmann (alexander.scherrmann@env.ethz.ch)

**Abstract.** Mediterranean cyclones are extratropical cyclones, typically of smaller size and weaker intensity than other cyclones that develop over the main open ocean storm tracks. Nevertheless, Mediterranean cyclones can attain high intensities, even comparable to the ones of tropical cyclones, and thus cause large socio-economic impacts in the densely populated coasts of the region. After cyclogenesis takes place, a large variety of processes are involved in the cyclone's development, contributing

with positive and negative potential vorticity (PV) changes to the lower-tropospheric PV anomalies in the cyclone center. Although the diabatic processes that produce these PV anomalies in Mediterranean cyclones are known in principle, it is still an open question whether they occur locally within the cyclone itself or remotely in the environment (e.g., near high orography) with a subsequent transport of high-PV air into the cyclone center. This study introduces a Lagrangian method to determine the origin of the lower-tropospheric PV anomaly, with an average amplitude of 1.2 PVU, relative to the tracks of cyclones identified

in ERA5 reanalyses. We define and quantify so-called "cyclonic" and "environmental" PV and find that the main part of the lower-tropospheric PV anomaly (60 %) is produced within the cyclone, shortly prior (−12 h) to the cyclones mature stage. Nevertheless, in 10 % of the cyclones the environmental PV production near the mountains surrounding the Mediterranean basin plays the dominant role for the low-tropospheric PV anomaly, and therefore the intensity of the circulation associated with these cyclones. An additional investigation of IFS simulations with detailed output from physical parameterizations reveals

that the major PV production inside the cyclone is typically due to convection and large-scale microphysics, whereas convection and turbulent momentum tendencies evoke most of the positive PV changes found in the environment.

## 1    Introduction

The Mediterranean is one of the most cyclogenetic region in the world (Trigo et al., 1999; Ulbrich et al., 2009; Neu et al., 2013). Mediterranean cyclones are prominent high-impact weather systems that have been increasingly analyzed in the last

years and thus, the driving processes of genesis and intensification of Mediterranean cyclones are fairly well known (Flaounas et al., 2022). They typically form in response to upper-tropospheric systems, such as potential vorticity (PV) cutoffs or narrow, meridionally-extended troughs that correspond to PV streamers (Appenzeller and Davies, 1992). These systems trigger baroclinic instability, which can provoke cyclogenesis (Massacand et al., 1998; Emanuel, 2005; Fita et al., 2006; Chaboureau et al., 2012; Flaounas et al., 2015; Raveh-Rubin and Flaounas, 2017).





Apart from the upper-level influence of the PV streamer, which corresponds to a southward deviation of the polar jet (Raveh-Rubin and Flaounas, 2017), diabatic processes in the lower troposphere usually contribute to the development and intensification of the cyclone (e.g., Campins et al. 2000; Horvath et al. 2006; McTaggart-Cowan et al. 2010) by generating a positive PV anomaly, e.g., due to latent heating by condensation in ascending air (Davis et al., 1993; Rossa et al., 2000; Čampa and Wernli, 2012). Thereby, the typically (very) strong cyclonic circulation in the lower troposphere in the mature stage of extratropical and

Mediterranean cyclones can be explained by the distinct vertical alignment of the diabatically produced positive PV anomaly at low levels with the upper-level PV streamer or cutoff, forming a so-called PV tower (Rossa et al., 2000; Čampa and Wernli, 2012).

PV was first introduced by Rossby (1939) and then defined by Ertel (1942) as:

$$\mathrm{PV} = \frac{1}{\rho}\boldsymbol{\eta} \cdot \boldsymbol{\nabla}\theta, \tag{1}$$

with air density $\rho$, the absolute vorticity vector $\boldsymbol{\eta}$, and potential temperature $\theta$. It is materially conserved under adiabatic conditions, such that only diabatic processes, e.g., latent heating and turbulence, can modify it (Ertel, 1942; Hoskins et al., 1985). PV can be inverted, given balance conditions, to recover the whole atmospheric state (e.g., Davis, 1992). Thereby, a positive PV anomaly induces a cyclonic circulation and thus makes PV an adequate variable to study cyclone dynamics. Considering the relevance of the upper and lower-tropospheric PV anomalies for the intensity of Mediterranean cyclones,

a recent study by Flaounas et al. (2021) found that for the most intense cyclones there is either a strong low-tropospheric diabatic PV anomaly or a very strong upper-tropospheric anomaly. For one third of intense Mediterranean cyclones the lower-tropospheric PV originates from deep convection located close to the cyclone center and the associated latent heating by condensation (Galanaki et al., 2016; Flaounas et al., 2018). However, a systematic assessment of the processes that produce PV in Mediterranean cyclones, also by other diabatic processes, such as turbulence and radiation, is currently missing. The recent

study by Attinger et al. (2021) showed that, in North Atlantic cyclones, also turbulent and radiative processes can contribute to the PV production in the center and along the fronts of the cyclone.

In some case studies of Mediterranean cyclones near the Alps, turbulence, particularly in the boundary layer, was found to create the lower-tropospheric PV (Flaounas et al., 2021). This PV originates from positive PV filaments, so-called PV banners (Aebischer and Schär, 1998; Rotunno et al., 1999; Epifanio and Durran, 2002; Schär et al., 2003; Flamant et al.,

2004), horizontally extending from the mountains to the Mediterranean sea. Indeed it has been shown by many studies that the complex topography surrounding the Mediterranean can lead to, e.g., Alpine lee cyclogenesis (Buzzi and Tibaldi, 1978; Buzzi et al., 2020). The lee side of the mountains is a favorable region for cyclogenesis, but it is unknown how many Mediterranean cyclones are directly affected by PV banners formed along the Alps or other mountain ranges (Flaounas et al., 2022), and how much the PV formed along the orography and then advected to the cyclone center contributes to the PV budget in the mature

stage of the cyclones.

To specifically analyze this process and, more generally, to quantify where and by what processes PV is produced that forms the low-tropospheric PV anomaly in mature Mediterranean cyclones, this study introduces a new method to identify whether low-tropospheric PV is produced "within" or "outside" the cyclone, which we refer to as "cyclonic" and "environmental" PV,





respectively. With this approach we quantify the relative contribution of processes in the environment and within the cyclone.
More specifically, our objectives are to identify:

1. Where and when, relative to the cyclone center and the mature stage of the cyclone, is the PV in the lower-tropospheric PV anomaly produced and what is the contribution of environmental PV production, particularly by topography?

2. Which diabatic processes predominantly produce PV within the cyclone and which in the environment and what is their relative contribution to the PV in lower troposphere at the mature stage of the cyclone?

The study is structured as follows: In Sect. 2 we provide an overview of the data and algorithms used, before introducing the concept of cyclonic and environmental PV in Sect. 3. The results for a large cyclone sample in ERA5 reanalyses are presented in Sect. 4 and 5. Finally, Sect. 6 provides results from a smaller set of cyclones in monthly IFS simulations, for which we quantify the PV modification by different diabatic processes. In the last section we summarize and discuss our results and give an outlook for future studies.

## 2 Data and methods

### 2.1 Datasets

In this study, we use two datasets, output from simulations with a special version of the ECMWF's integrated forecasting system (IFS) and ERA5 reanalysis data from 1979 to 2020 (Hersbach et al., 2020). From ERA5, we use wind and PV fields (PV calculated on model levels), regridded to a 0.5° x 0.5° grid on 98 vertical model levels. We also use output from twelve monthly
IFS simulations, which were obtained using cycle 43R1 of the hydrostatic IFS model (Roberts et al., 2018), as described in Attinger et al. (2021). Each simulation covers 35 days and is initialized at the beginning of each month from December 2017 to November 2018. The available data covers the northern hemisphere on a 0.4° x 0.4° grid on 83 vertical model levels. The special IFS version provides detailed output of temperature and momentum tendencies from all parameterizations, from which we can calculate the corresponding PV tendencies enabling a process-based investigation of the formation of PV anomalies
along air parcel trajectories. Here we follow the approach by Joos and Wernli (2012) which was later used, e.g., by Attinger et al. (2019, 2021) and Spreitzer et al. (2019). Table 1 summarizes the processes and corresponding PV tendencies considered in our study. Details about the parameterizations, as well as the calculation of the PV tendencies can be found in Attinger et al. (2019).

### 2.2 Cyclone tracking and characteristics

For cyclones in ERA5, we applied the cyclone tracking method of Wernli and Schwierz (2006) with the modifications described in Sprenger et al. (2017), which focuses on local minima in the sea level pressure (SLP) field. For the IFS, the tracks were identified with the approach by Attinger (2020) who adapted the method of Wernli and Schwierz (2006) by further including fields of relative vorticity and equivalent potential temperature to improve the identification of cyclones and their frontal

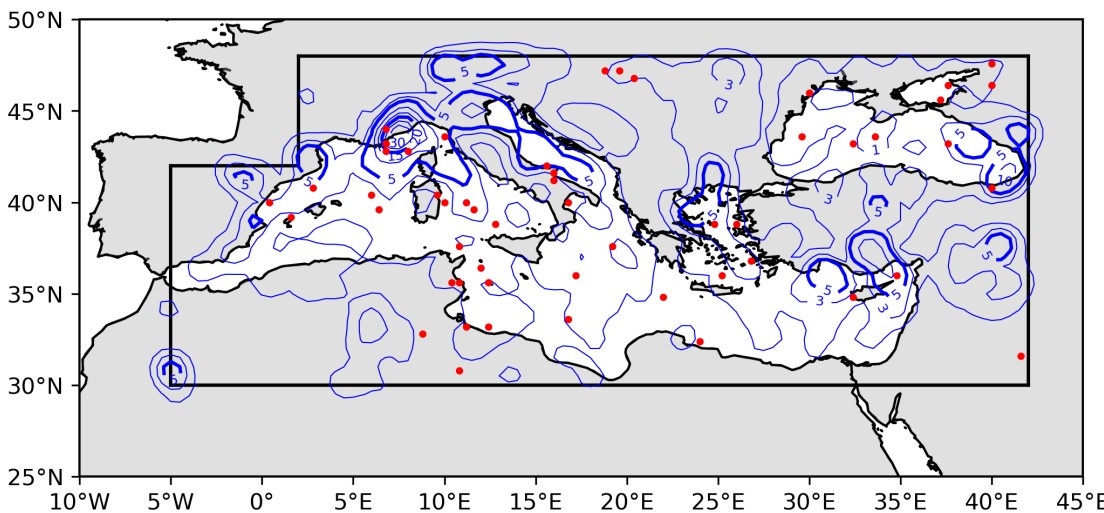

**Figure 1.** Location of all considered Mediterranean cyclones at their mature stage in ERA5 (blue contours) and in the IFS simulations (red dots). The blue contours show the number of cyclones at the specific location (averaged in a 1.5° x 1.5° box around the grid points) with contour levels of 1, 3, 5, 10, 15, 20, 25, 30 (level 5 and 20 contours are bold).

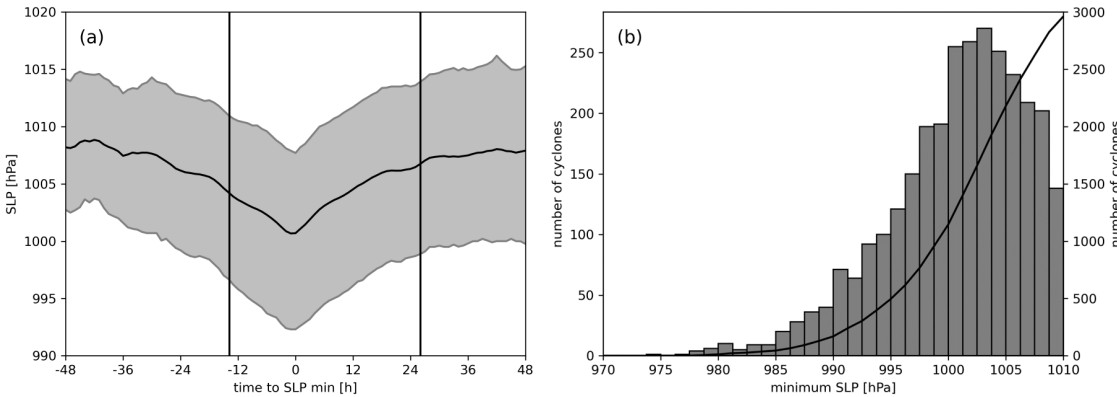

**Figure 2.** SLP statistics for almost 3000 Mediterranean cyclones in ERA5. (a) Time evolution of minimum SLP along cyclone tracks; the black line shows the mean, and grey shading the range between the 10[th] and 90[th] percentiles; vertical lines indicate the average time of cyclogenesis and cyclolysis; (b) Histogram of minimum SLP along each cyclone track (bin-width of 1.25 hPa), and cumulative number of cyclones (black line).



**Table 1.** Physical processes we consider in the IFS simulations.

| Abbreviation | Process | Tendencies |
|---|---|---|
| $\mathrm{PVR_{conv}}$ | Convection | $\dot{\theta}_{\mathrm{conv}}$, $\boldsymbol{F}_{\mathrm{conv}}$ |
| $\mathrm{PVR_{turb}}$ | Turbulence and gravity wave drag (orographic and non-orographic) | $\dot{\theta}_{\mathrm{turb}}$, $\boldsymbol{F}_{\mathrm{turb}}$ |
| $\mathrm{PVR_{ls}}$ | Large-scale microphysics | $\dot{\theta}_{\mathrm{ls}}$ |
| $\mathrm{PVR_{rad}}$ | Long- and short-wave radiation | $\dot{\theta}_{\mathrm{rad}}$ |

structures. We define a cyclone as a Mediterranean cyclone if it reaches its mature stage in the region indicated by the black

box in Fig.1. The mature stage is defined as the time of maximum relative vorticity at $850\,\mathrm{hPa}$ along the IFS tracks, and as the time of minimum SLP along the ERA5 tracks. For ERA5, in order to focus on the most intense cyclones, we only retain cyclones with a SLP minimum below $1010\,\mathrm{hPa}$ and have at least a $5\,\mathrm{hPa}$ difference between its minimum and maximum SLP value along the track. Finally, we select cyclones with a sufficiently large low-level PV anomaly, i.e., such that we can initialize at least 200 backward trajectories from their mature stage (see below). Applying all these criteria resulted in 55 cyclone tracks

in the twelve monthly IFS simulations (red dots in Fig. 1) and 2969 tracks in ERA5 (blue contours in Fig. 1). Similar to, e.g., Campins et al. (2011), we find the main cyclone hot spots in the Gulf of Genoa, the Adriatic Sea, close to Cyprus, and near the Atlas mountains.

Figure 2a shows the average time evolution of SLP along the ERA5 cyclone tracks, centered at the time of the mature stage $(0\,\mathrm{h})$. The average lifetime of of these cyclones is about $36\,\mathrm{h}$, indicated in Fig. 2a by the black vertical lines. The first track point

is identified on average about $13\,\mathrm{h}$ before the mature stage $(t = -13\,\mathrm{h})$ and the last track point about $24\,\mathrm{h}$ after the mature stage $(24\,\mathrm{h})$. The mean SLP evolution gradually decreases during the two days prior the mature stage from about 1009 to $1002\,\mathrm{hPa}$, and an almost symmetric increase in the next two days. The gray shading in Fig. 2 shows that the SLP evolution of individual cyclones is more variable. The SLP evolution of IFS cyclones is similar with a slightly lower average of $998\,\mathrm{hPa}$ at $t = 0\,\mathrm{h}$ compared to $1002\,\mathrm{hPa}$ in ERA5, and a slightly shorter average lifetime of $33\,\mathrm{h}$ (not shown). The distribution of minimum SLP

along ERA5 cyclone tracks is shown in Fig. 2b. The majority of cyclones $(66\,\%)$ have a minimum SLP larger than $998.75\,\mathrm{hPa}$, indicated by the cumulative number of cyclones (black line). The largest number of cyclones is found with a SLP minimum of about $1002.5\,\mathrm{hPa}$. The distribution is strongly skewed with the deepest cyclones reaching minimum SLP values slightly lower than $980\,\mathrm{hPa}$. Due to the significantly lower number of cyclones in the IFS simulations, the corresponding histogram does not provide statistically significant insights into the SLP distribution in the IFS simulations. The values in Fig. 2 are in

accordance with a recent study by Aragão and Porcù (2022), who quantified the cyclone activity in the Mediterranean using ERA5. Considering seasonality, we find the largest number of cyclones in winter (DJF) and about half that number during summer (JJA), representing a seasonal cycle similar to the one presented in Flaounas et al. (2015).





### 2.3 Lagrangian perspective on diabatic PV modification

In order to investigate the processes that contributed to the formation of the lower-tropospheric PV anomalies in the identified
IFS cyclones, we first calculate backward trajectories from the lower-tropospheric PV anomaly in each cyclone and then
accumulate local PV tendencies along trajectories. In addition, we keep track of where the anomalies formed that eventually
are located in the cyclone center. It is important to note that such an analysis requires the availability of 3-dimensional diabatic
tendency fields due to individual processes, which are not routinely available (e.g. in ERA5). Therefore, this Lagrangian
analysis could only be applied to the much smaller set of IFS cyclones and not to cyclones in ERA5.

The PV tendency $\mathrm{PVR}_i$, due to a diabatic process $i$, is quantified by the PV tendency equation (Ertel, 1942):

$$\mathrm{PVR}_i = \frac{1}{\rho}(\boldsymbol{\eta} \cdot \boldsymbol{\nabla}\dot{\theta}_i + \boldsymbol{\nabla} \times \mathbf{F}_i \cdot \boldsymbol{\nabla}\theta), \tag{2}$$

with the potential temperature $\left(\dot{\theta}_i = \mathrm{D}\theta_i/\mathrm{D}t\right)$ and momentum tendencies ($\mathbf{F}_i$), due to the diabatic process $i$ (see Table 1).
To quantify the contribution of each process to the lower-tropospheric PV, we follow the approach of Crezee et al. (2017),
Spreitzer et al. (2019), and Attinger et al. (2019), in which the integration of $\mathrm{PVR}_i$ along the trajectory $\boldsymbol{r}(t)$ from time $t_n$ to
$t_0$ ($t_n < t_0$) yields the accumulated PV ($\mathrm{APV}_i$) of the particular process between times $t_n$ and $t_0$. Here, $t_n$ indicates the end
time of the backward trajectory calculation initialized at $t_0$ and the accumulated PV tendency $\mathrm{APV}_i$ is approximated by the
summation of the hourly PV tendencies along the trajectory:

$$\mathrm{APV}_i\left(\boldsymbol{r}(t_0), t_n\right) \approx \sum_{k=1}^{n} \frac{\mathrm{PVR}_i\left(\boldsymbol{r}(t_k), t_k\right) + \mathrm{PVR}_i\left(\boldsymbol{r}(t_{k-1}), t_{k-1}\right)}{2}\Delta t, \tag{3}$$

Here, we use the average PV tendency of two consecutive trajectory points, to account for temporal variations of the processes
between hourly output times. However, this approach does not yield a perfectly closed PV budget and hence the time integration
results in a residual, whose potential sources were explained in detail in Attinger et al. (2019). For the Mediterranean IFS
cyclones, we find an average residual of $-0.2\,\mathrm{PVU}$ ($1\,\mathrm{PVU} = 10^{-6}\,\mathrm{m^2\,s^{-1}\,K\,kg^{-1}}$), which is expected to have a negligible
effect on cyclone dynamics.

Similar to the PV modifications due to explicitly accumulated PV tendencies, $\mathrm{APV}_i$ (Eq. 3), we define the total PV change as
follows:

$$\mathrm{apv}(t_0, t_n) = \sum_{k=1}^{n} \Delta\mathrm{PV}(t_k) = \sum_{k=1}^{n}\left[\mathrm{PV}(t_{k-1}) - \mathrm{PV}(t_k)\right]$$
$$= \mathrm{PV}(t_0) - \mathrm{PV}(t_n). \tag{4}$$

Flaounas et al. (2021) showed that most of the diabatic processes that are relevant for Mediterranean cyclone dynamics occur
close to the cyclone center and the low-level PV anomaly extends to about $150\,\mathrm{km}$ from the cyclone center. Therefore, we
initialize trajectories within a distance of $200\,\mathrm{km}$ around the SLP minimum (dashed ring in Fig. 3a). We restrict the initialization
to the lower troposphere between $975\text{-}700\,\mathrm{hPa}$, to focus on the lower-tropospheric PV anomaly that is typically largest in this



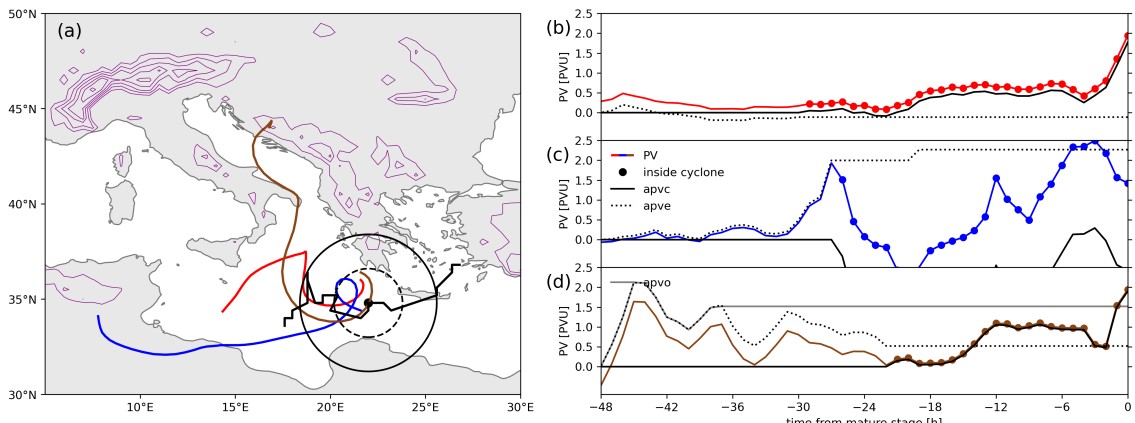

**Figure 3.** Application of the method to quantify cyclonic vs. environmental PV production illustrated for 3 example trajectories. (a) Path of the 48 h backward trajectories (colors), the cyclone track (black), and the location of the mature stage (black dot). The region of trajectory initialization and the cyclone effective radius are highlighted by the dashed and solid black circle, respectively. The purple contours show topography, starting at 800 m with 400 m steps; (b-d) PV evolution (color) along the different trajectories and the diagnosed cyclonic (solid black line), environmental (dotted black line), and orographic PV (grey line, only in (d)).

layer in extratropical cyclones (Čampa and Wernli, 2012). Considering the climatological PV value over the Mediterranean Sea between 975-700 hPa, which is around 0.36 PVU, trajectories are finally initialized from the grid points within the circle if PV exceeds 0.75 PVU, which is about twice the climatological average value and thus a good threshold for a lower-tropospheric positive PV anomaly.

We compute backward trajectories using LAGRANTO (Wernli and Davies, 1997; Sprenger and Wernli, 2015) until 48 h prior to the mature stage of the cyclone ($-48$ h), which is an adequate time period considering the short life span of Mediterranean cyclones (Trigo et al., 1999; Aragão and Porcù, 2022). Along these, we trace surface pressure, PV, and in the IFS simulations the available PV rates due to different diabatic processes. This allows us to define "cyclonic" and "environmental PV" as explained in the next section.

## 3 Concept of cyclonic and environmental PV

### 3.1 Methodology

To motivate the concept of distinguishing between cyclonic and environmental PV and to illustrate our approach, Fig. 3a shows 3 example trajectories for one of the IFS cyclones. For the trajectory that is located always over the sea (red), we find

a strong increase in PV within 6 hours before the mature stage (Fig. 3b), whereas the trajectories from northern Africa (blue) and the Dinaric Alps (brown) experience both strong increases and decreases in PV at earlier times. However, it is a priori not known whether these PV modifications occurred close to the propagating cyclone or remotely, i.e., in the environment of



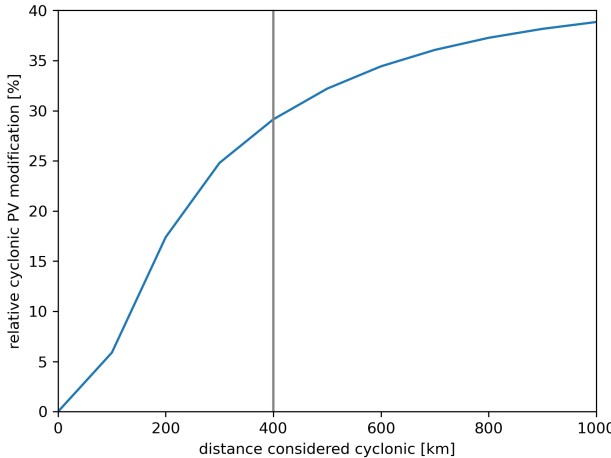

**Figure 4.** Average percentage of total PV modification attributed as cyclonic PV for different cyclone radii, along all trajectories of ERA5 cyclones.

the cyclone. Our method identifies PV changes as either "cyclonic" (PV produced inside the cyclone) or "environmental" (PV produced outside the cyclone, which we refer to as "in the environment"). To investigate the relative importance of cyclonic and environmental PV on the cyclone, we compare their strength to the total PV observed at the time of the mature stage. Hereafter, we refer to this PV value as PVM.

To distinguish between cyclonic an environmental PV we rely on: (i) the cyclone track at times prior to the mature stage, (ii) backward trajectories from the high PV regions in the cyclone center of the mature stage, and (iii) a criterion to define whether the trajectories are inside the cyclone or outside. For (iii), we pragmatically define a "cyclone effective area" as a circle with a radius yet to be determined around the cyclone center. All PV modifications taking place within this circle are considered to be related to the cyclone dynamics (i.e., "inside the cyclone"), while those outside the circle are considered as "environmental".

To better understand how the size of the cyclone effective area affects our results, we used different radii of the circle to quantify cyclonic vs. environmental PV production. Figure 4 shows the percentage of the average absolute PV modifications along the backward trajectories that occur within the circle around the cyclone center, for all ERA5 cyclones. This percentage strongly increases to about 28 % when increasing the radius up to 400 km. For larger radii the increase flattens reaching 38 % cyclonic PV change for a radius of 1000 km. Considering also the typical size of Mediterranean cyclones (e.g., Trigo et al., 1999; Campins et al., 2011) we thus consider a radius of 400 km as a good choice for the cyclone effective radius.

We therefore define segments of trajectories as "inside the cyclone", if the distance between the trajectory's position at time $t$, $\boldsymbol{r}(t)$, and the current position of the cyclone center, $\boldsymbol{C}(t)$, is smaller than 400 km (black circle in Fig. 3a) and else as "in the environment". Note that this criterion is dynamic in the sense that the cyclone effective area (with fixed radius) is moving along the cyclone track. Thus, by design, at $t = 0\,\mathrm{h}$ all PV changes are cyclonic since trajectories are always initialized within 200 km from the cyclone center. In contrast, at times prior to cyclogenesis, i.e., prior to the first cyclone track point, all PV





modifications are considered as environmental. The net cyclonic PV change (apvc, solid black lines in Fig. 3b-d) and the environmental PV change (apve, dotted black lines) can be quantified similar to Eq. (4) by summing the PV modifications at

times when the trajectory is inside the cyclone (colored dots in Fig. 3b-d) and in the environment, respectively. Note that both apvc and apve are intialized to zero at $t = -48\,\mathrm{h}$ and then integrated forward in time. The sum of apvc and apve results in the total PV change apv.

For the trajectory in Fig. 3b, our analysis reveals that the steep increase in PV in the last $6\,\mathrm{h}$ occurs within the cyclone and the cyclonic PV contributes with $+1.8\,\mathrm{PVU}$ to the $2\,\mathrm{PVU}$ PVM. In contrast the environment provides a negative PV contribution

of $-0.15\,\mathrm{PVU}$, which results in a net total PV change of $+1.65\,\mathrm{PVU}$, which together with an initial value of $0.35\,\mathrm{PVU}$ at $t = -48\,\mathrm{h}$ (hereafter PVS) yields the $2\,\mathrm{PVU}$ PVM. In comparison, for the trajectory in Fig. 3c the environment produces PV up to $t = -27\,\mathrm{h}$ after which the cyclone both produces and destroys PV resulting in a net PV destruction in the cyclone, such that only environmental PV contributes positively to the PVM of $1.5\,\mathrm{PVU}$ in this case.

Our study further aims to quantify the role of PV production near mountains for Mediterranean cyclones and thus we define

"orographic PV changes" (apvo, grey line in Fig. 3d) according to the following pragmatic criteria: (i) the trajectory passes over orography higher than $800\,\mathrm{m}$, and (ii) the difference between surface pressure and the trajectory pressure is less than $100\,\mathrm{hPa}$ (i.e., the trajectory is close to the surface). To give an example, the trajectory in Fig. 3d experiences a steep early increase of environmental PV when it is located over the Dinaric Alps (purple contours in Fig. 3a). In this case, the early PV increase (after $t = -48\,\mathrm{h}$) is likely related to orographic PV modification (grey line). In this case, we refer to the remaining

$0.5\,\mathrm{PVU}$ apve at $t = 0\,\mathrm{h}$ as being produced by the flow near the mountains, which is the relevant PV for the cyclone dynamics at the mature stage, the focus of our study. It is $0.5\,\mathrm{PVU}$ of orographic PV and not $1.5\,\mathrm{PVU}$, because the environmental PV at $t = -38\,\mathrm{h}$, which is completely orographic (dotted and grey lines), is reduced by the environment between $t = -38\,\mathrm{h}$ and $t = -22\,\mathrm{h}$ in absence of orography. Therefore, the environmental PV consists of $+1.5\,\mathrm{PVU}$ early orographic PV and $-1\,\mathrm{PVU}$ non-orographic environmental PV, which results in $+0.5\,\mathrm{PVU}$ orographic PV at the mature stage. It is important to note that

with this approach, the "orographic PV" refers to the PV modification near mountains without identifying or specifying the underlying physical process. This PV modification can be due to frictional effects and/or cloud-related diabatic processes.

## 3.2 Exemplary application to cyclone Zorbas

To provide an example, we quantify environmental and cyclonic PV modifications of medicane Zorbas (Portmann et al., 2020). Zorbas reached its minimum SLP of $992\,\mathrm{hPa}$ at 0400 UTC on 28 September 2018 (black dot in Fig. 5a). Figure 5a also shows

the track of Zorbas (black line), together with the $48\,\mathrm{h}$ backward trajectories initialized from the core of Zorbas at the time of minimum SLP. The hourly trajectory segments are colored according to the PV modification within this hour. There is one major airstream from the Black Sea across the Aegean Sea towards Zorbas. Most trajectories show a strong PV increase close to the location of Zorbas and sporadic PV changes of both signs further upstream. Similarly to Fig. 3b-d but now averaged over all trajectories shown in Fig. 5a, Fig. 5b shows the evolution of PV and of the diagnosed cyclonic and environmental

PV. Trajectories have an average PVS of $0.4\,\mathrm{PVU}$ at $t = -48\,\mathrm{h}$ and an average PVM of $1.5\,\mathrm{PVU}$ at $t = 0\,\mathrm{h}$. This leads to an average PV gain of $1.1\,\mathrm{PVU}$ along the trajectories. At early times ($t < -27\,\mathrm{h}$), i.e., before the genesis of Zorbas, averaged PV



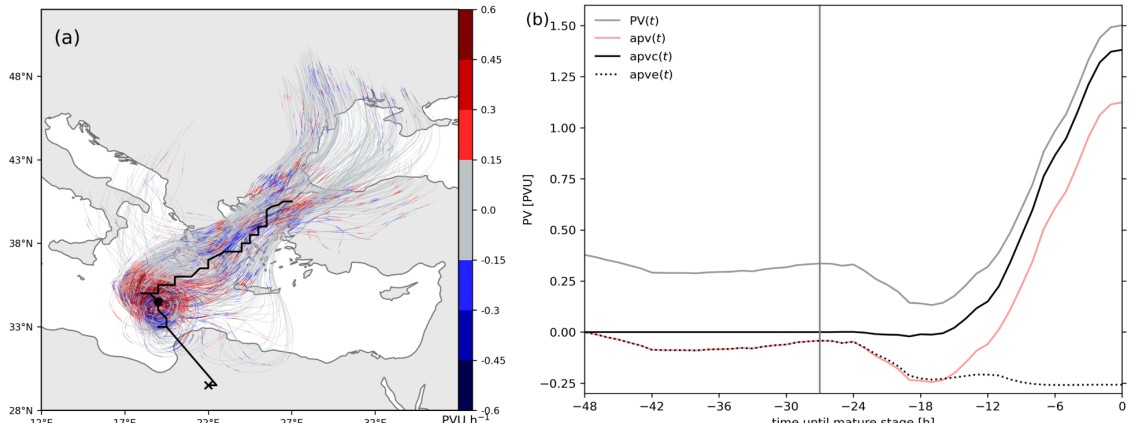

**Figure 5.** Case study for cyclone Zorbas in the mature stage at 0500 UTC on 28 September 2018 ($t = 0$ h). (a) Backward trajectories from the low-tropospheric PV anomaly within Zorbas, colored with hourly PV changes (colors, in $\mathrm{PVU\,h^{-1}}$); the black dot indicates the location of the cyclone center, the black line the cyclone track from genesis over Libya marked with a black cross. (b) Time evolution of the average PV along the trajectories shown in (a) (grey), of the accumulated PV change, apv, starting at $t = -48$ h (red), and of the accumulated cyclonic (black solid) and environmental (black dotted) PV modification, respectively. The time of cyclogenesis at $t = -27$ h is shown by the vertical grey line.

decreased by a bit less than $0.1$ PVU due to environmental PV destruction (negative apve, dotted line). After cyclogenesis until $t = -16$ h, cyclonic PV remains close to zero indicating that no PV modification occurred in the vicinity of the track of Zorbas (at a distance smaller than $400$ km from the center). In this period, apve becomes even more negative (about $-0.25$ PVU).

After $t = -16$ h almost all trajectories enter the effective cyclone area. Therefore apve($t$) remains constant after $t = -16$ h and the very strong PV increase up to $1.5$ PVU is entirely due to cyclonic PV production. This is what we would expect for a tropical-like cyclone that is predominantly driven by (deep) convection and intense low-level PV production close to its core. In essence, Fig. 5b shows that the PV gain of $1.1$ PVU results from a cyclonic PV production of $1.35$ PVU and an environmental PV destruction of $-0.25$ PVU.

**4   Cyclonic and environmental PV in ERA5**

In this section we apply our methodology to all Mediterranean cyclones identified in ERA5. Figure 6a shows the time evolution of PV averaged along all trajectories from all ERA5 cyclones, where $t = 0$ h again corresponds to the mature stage of the cyclones. The grey shaded area indicates the range between the 10th and 90th percentiles. From $t = -48$ h until $t = -12$ h, PV is on average fairly well conserved. In the last $12$ h before the mature stage, PV increases to $1.5$ PVU, resulting in a PV

gain of about $1$ PVU. Therefore, the PV anomaly that defines the cyclone in the lower troposphere mainly forms within 12 hours prior to the mature stage. It is noteworthy that some trajectories already start with PV $\geq 1$ PVU. These high PVS values originate from PV production processes prior to $t = -48$ h. In most cases, they were produced earlier than cyclogenesis and



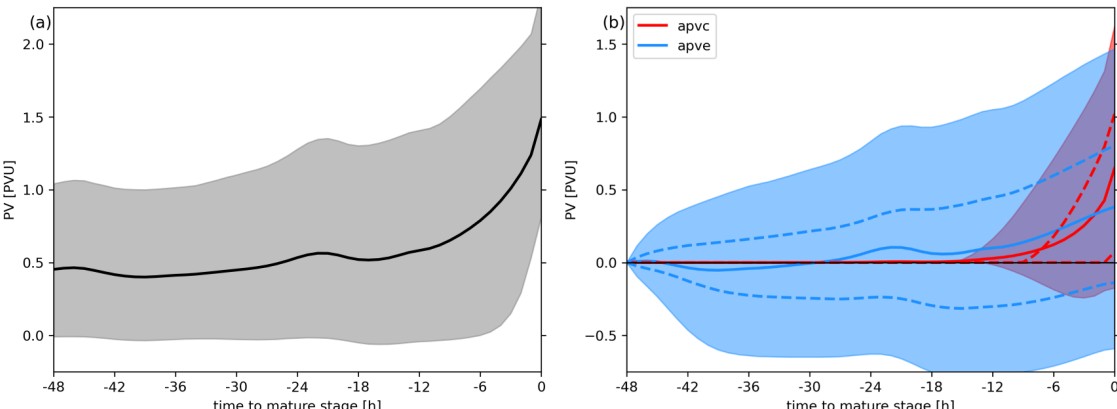

**Figure 6.** Time evolution of PV along backward trajectories initialized from all considered ERA5 cyclones: (a) average PV (black line) and the $10^{\text{th}}$ and $90^{\text{th}}$ percentiles (grey shading); (b) cyclonic (red) and environmental (blue) PV modification (solid lines show the mean, dashed lines the $25^{\text{th}}$ and $75^{\text{th}}$, and the shading the $10^{\text{th}}$ and $90^{\text{th}}$ percentiles.

are therefore environmental. Figure 6b shows the time evolution of the cyclonic (apvc; red) and environmental (apve; blue) PV modifications. It reveals that most of the PV modification occurs inside the cyclone (mean value of 0.65 PVU) during the 12 h

prior to the mature stage. Cyclonic PV starts increasing around $t = -13$ h, matching the average time of cyclogenesis (Fig. 2a). In contrast, the contribution of environmental PV is about half as strong with a mean of 0.38 PVU. Interestingly, until about $t = -15$ h, the mean environmental PV modification is close to zero indicating that positive and negative contributions cancel each other. However, environmental PV shows also a particularly large case-to-case variability (see shading). For instance, total apve at $t = 0$ h ranges from $-0.55$ to 1.5 PVU ($10^{\text{th}}$ and $90^{\text{th}}$ percentiles) compared to apvc with a range from $-0.2$ to

1.6 PVU.

Next, we calculate for each cyclone the percentage of PVM that is produced inside the cyclone (apvc(0 h)/PVM), in the environment (apve(0 h)/PVM), and how much is due to PVS (PVS/PVM). On average, the contributions from the cyclone and PVS are the largest with 40 % and 37 %, respectively, whereas the environment provides a contribution of 23 % to PVM. Throughout the year these percentages remain fairly constant (not shown). These findings suggest that the lower-tropospheric

PV is mainly produced by processes in the vicinity of the cyclone center; however, contributions from environmental PV are non-negligible and can dominate for individual cyclones, as further discussed below.

Up to this point we focused on the total PVM in the lower troposphere. However, the PVM involves the climatological background value of PV and the cyclone-related PV anomaly. To quantify the strength of the lower-tropospheric PV anomaly (PV$^\star$), we thus subtract the climatological PV (PV$_{\text{c}}$) from PVM and PVS, i.e., PVM$^\star$ = PVM − PV$_{\text{c}}$ and PVS$^\star$ = PVS − PV$_{\text{c}}$. Table

2 lists the climatological PV values for each month and the entire year. These values correspond to the average PV in ERA5 during 1979-2020, between 975-700 hPa over ocean grid points within the region we regard as the Mediterranean (Fig. 1), while excluding grid points within cyclone masks. The values show only a weak seasonal cycle with a minor peak in summer (July, August).





**Table 2.** Climatological PV values in the lower troposphere (975-700 hPa) over the Mediterranean Sea, calculated from ERA5 from 1979-2020, excluding cyclone masks, see text for details.

|  | Year | Jan | Feb | Mar | Apr | May | Jun | Jul | Aug | Sep | Oct | Nov | Dec |
|---|---|---|---|---|---|---|---|---|---|---|---|---|---|
| $PV_c$ [PVU] | 0.357 | 0.344 | 0.330 | 0.336 | 0.341 | 0.363 | 0.379 | 0.403 | 0.393 | 0.363 | 0.348 | 0.337 | 0.344 |

Focusing now on the PV anomaly, the contributions change to 56 % inside the cyclone, 32 % in the environment and 12 %
due to the initial PV anomaly (PVS⋆). By design the ratio of the cyclonic and environmental contributions is the same when considering the total lower-tropospheric PV or the PV anomaly. Considering these values for individual cyclones, we find that 59.2 % of the cyclones produce at least 50 % of their PV anomaly inside the cyclone. Nevertheless, 1100 cyclones (37.3 %) have at least 50 % of PV⋆ produced in the environment, and 200 cyclones (7 %) have an anomalously high PVS⋆ that accounts for more than 50 % of PV⋆. It is important to note that a cyclone can, e.g., have 50 % of PV⋆ produced inside the cyclone as
well as 50 % of PV⋆ produced in the environment, if PVS⋆ is negative, i.e., initial PV values are smaller than the climatological ones. In the next section we systematically identify regions where large PV changes occur that eventually end up in the center of Mediterranean cyclones and further quantify the role of the mountains in environmental PV production.

## 5 Regions of large PV changes and orographic influences

### 5.1 Climatological analysis

The previous results showed that most of the PV changes occur inside the cyclone, but yet a significant amount of PV is produced in the environment. Here, we use the trajectory data from all cyclones in ERA5 to identify distinct regions in the Mediterranean that systematically provide large PV changes to Mediterranean cyclones. Therefore, Fig. 7 shows the absolute frequency of large PV changes greater or equal to $0.15\,\mathrm{PVU\,h^{-1}}$ (positive changes in Fig. 7a,b; negative changes in Fig. 7c,d), within the cyclone (Fig. 7a,c) and in the environment (Fig. 7b,d). The figure indicates that large PV changes (positive
and negative) less frequent in cyclones than in the environment. However, especially over the Mediterranean Sea, large PV production predominantly occurs within the cyclones (Fig. 7a). Furthermore, inside the cyclone positive PV modifications are significantly more frequent than negative ones (Fig. 7a,c). Thus they result in an overall strong PV gain inside the cyclone, as shown in Fig. 6b. In contrast, for trajectories in the environment there are only a few areas over land, where PV is significantly more frequently produced than destroyed. Hence, in the environment, PV is almost evenly frequently produced and destroyed,
however, on average production is stronger and thus explains the fluctuations of apve($t$) and its only modest increase with time (Fig. 6b). It is important to note, that near the Atlas mountains large PV changes in the cyclone are relatively speaking as frequent as over the western Alps, i.e., when trajectories pass over the western Alps or the Atlas mountains they are evenly likely to gain $\geq 0.15\,\mathrm{PVU}$ (not shown), as the diabatic processes near the mountains are similar atall locations. However, there are fewer cyclones over Morocco and Algeria and thus the absolute PV production frequencies are relatively low (Fig. 7a).



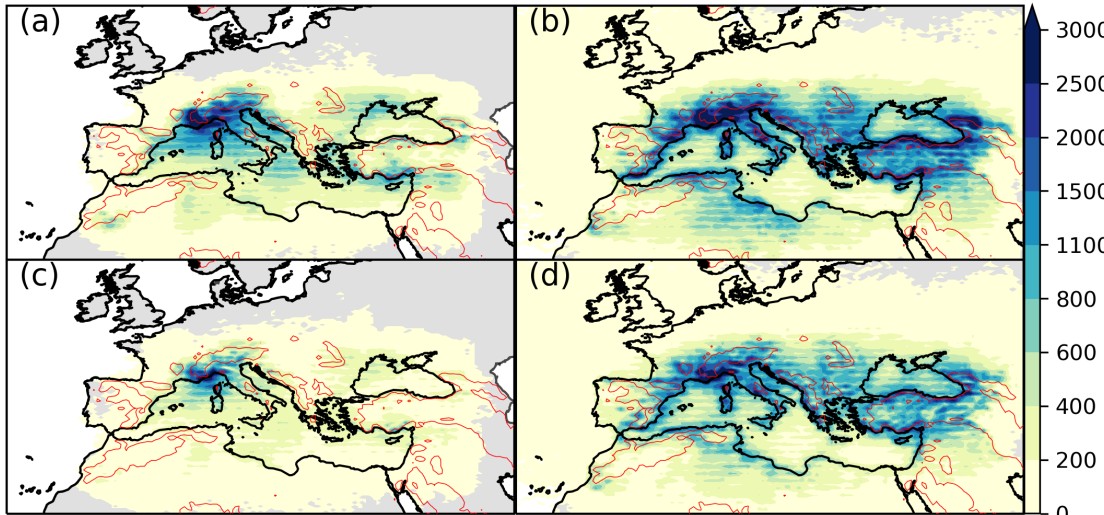

**Figure 7.** Absolute frequencies of hourly PV modifications $\geq 0.15\,\mathrm{PVU\,h^{-1}}$ within the cyclone (a) and in the environment (b) and of hourly PV modifications $\leq -0.15\,\mathrm{PVU\,h^{-1}}$ within the cyclone (c) and in the environment (d). The red contour shows orography above $800\,\mathrm{m}$.

As large PV changes are most frequent in the vicinity of the mountains, we quantify the contribution of orographic PV (Sect. 3.1) to the lower-tropospheric PV anomaly. To specify the role of the mountains, we thus extract all "strongly orographic" cyclones, defined as cyclones dominated by environmental PV for which the orographic contribution provides at least $25\,\%$ of the PV anomaly. This definition yields 310 ($10.4\,\%$) strongly orographic cyclones with hot spots in the gulf of Genoa and the eastern region of the Black Sea (blue contours in Fig. 8). In the Gulf of Genoa, PV banners from the western Alps and the

Mistral wind might provide high PV to intensifying cyclones, whereas in the eastern Black Sea the cyclones are surrounded by the Caucasus and Taurus mountains. The location of most of the strongly orographic cyclones close to the mountains suggests that it is rather unlikely that mountains also strongly impact lower-tropospheric PV in cyclones that reach their mature stage over the central Mediterranean Sea. To provide an overview of the dynamics of strongly orographic cyclones, the next section provides two case studies.

**5.2    Two case studies of cyclones with large orographic PV contributions**

We consider a cyclone that formed near Marseilles at 1300 UTC on 12 December 1988, with a central SLP of $1013.4\,\mathrm{hPa}$, which moved south (black line in Fig. 9a) to the Mediterranean Sea, where it reached its mature stage at 0400 UTC on 13 December 1988 with a minimum SLP of $1008.7\,\mathrm{hPa}$. It gained most of its lower-tropospheric PV due to interaction with orography. Figure 9a shows the backward trajectories from the cyclone center in the mature stage, colored according to hourly

PV changes. There is one major airstream moving from western Europe along the western Alps into the cyclone center. Many trajectories gain a lot of PV in the Rhone valley when flowing along the western edge of the Alps. Figure 9b shows the time

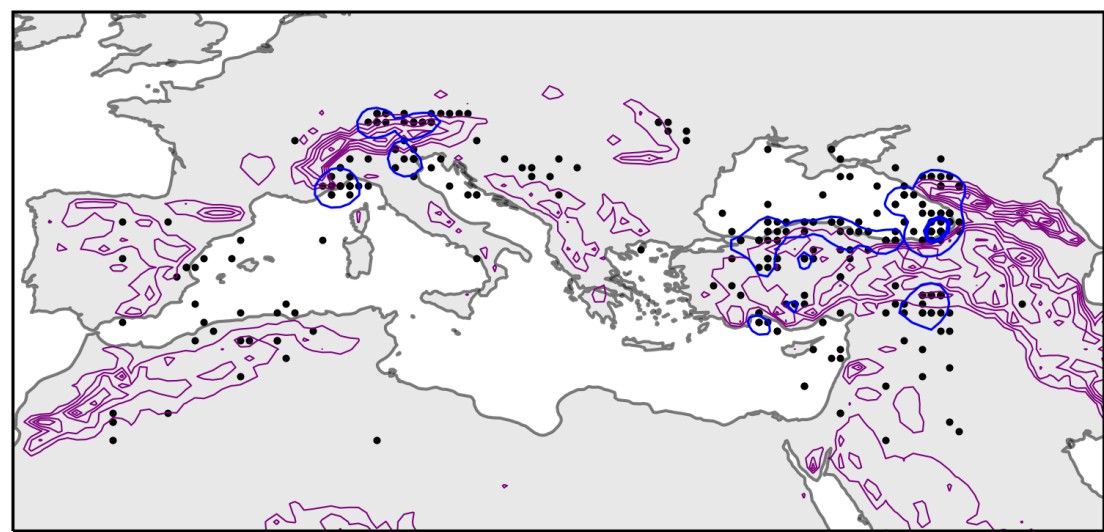

**Figure 8.** Location of the mature stage of so-called orographic cyclones (black dots). Contours mark the hotspots with 1 and 5 cyclones (averaged in a 1.5° x 1.5° box around the grid points). Purple contours show orography higher than 800 m with 400 m intervals.

evolution of PV (grey) and the accumulated PV modification (red). PV on average doubles within 48 h reaching a PVM of 1.25 PVU with an initial PVS of approximately 0.6 PVU. The accumulated PV (apv) is split in its cyclonic (black solid) and environmental (black dotted) parts. This reveals that the strong increase in PV between $t = -36\,\mathrm{h}$ and $t = -21\,\mathrm{h}$ occurs in

the environment. After the trajectories enter the cyclone effective radius (vertical grey line), PV remains fairly constant due to an approximately zero net PV change inside the cyclone. Thus the PV increase of 0.65 PVU almost entirely originates from the environment (75 % of PV⋆), whereas the remaining 25 % are provided by an anomalously high PVS. Quantifying the orographic PV contribution (black dashed line in Fig. 9b), we find that the environmental PV was mainly produced near the Alps.

A second cyclone formed in the northern Aegean Sea at 0000 UTC on 31 January 1988 and then crossed northwestern Turkey to enter the Black Sea (Fig. 10a), where at 1400 UTC on 31 January 1988 it reached its mature stage with a central SLP of 980.9 hPa (black dot in Fig. 10a). Three major airstreams fed into the cyclone, one from North Africa, one from the Mediterranean Sea, and one from the eastern parts of the Black Sea. All of them had strong PV increases when they crossed the underlying orography (Fig. 10a). Figure 10b shows the time evolution of PV (grey) and of accumulated PV (red). PV

on average quintuples within 48 h reaching a PVM of 1.5 PVU with an initial PVS of 0.3 PVU. The overall increase in PV originates from environmental PV modifications (dotted black line in Fig. 10), whereas the diabatic processes inside the cyclone destroy PV (solid black line in Fig. 10). For this cyclone the environmental PV modification produced almost the entire PV anomaly, of which 60 % was produced by interaction with the mountains in all three airstreams (black dashed line in Fig. 10).


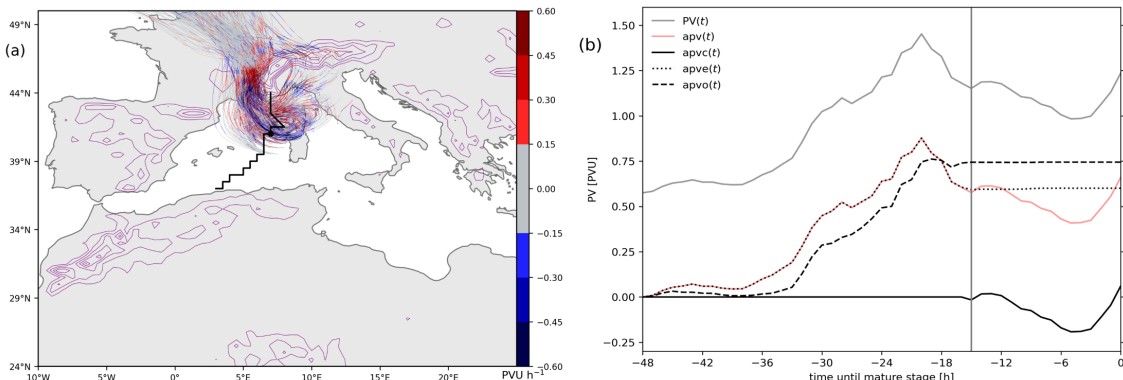

**Figure 9.** Same as Fig. 5 for a cyclone with its mature stage at 1300 UTC on 12 December 1988. Orographic PV modification is shown by the black dashed line.

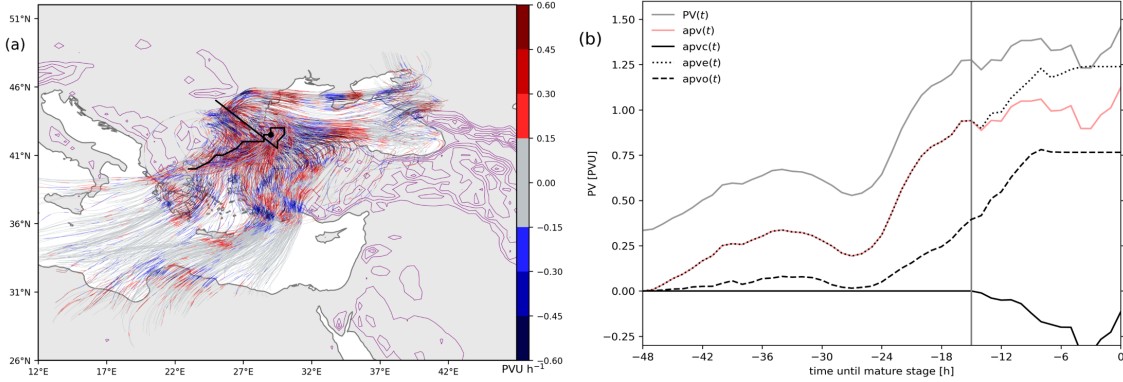

**Figure 10.** Same as Fig. 9 for a cyclone that had its mature stage at 1400 UTC on 31 January 1988.

Comparing these two cyclones, we find the main difference in the minimum SLP, the number of trajectories, and thereby the
volume of the lower-tropospheric PV anomaly, which might explain the significantly lower SLP of the cyclone in the Black Sea.

These case studies provide new insights with regard of the role of PV production near orography for cyclone intensification. In contrast to the majority of cyclones with diabatic PV production inside the cyclone, the PV of these cyclones is produced in the environment, i.e., by interaction with orography, and then advected into the cyclone. However, with our ERA5 analysis, we
cannot identify the underlying diabatic processes responsible for the PV production, which we address in the next section.

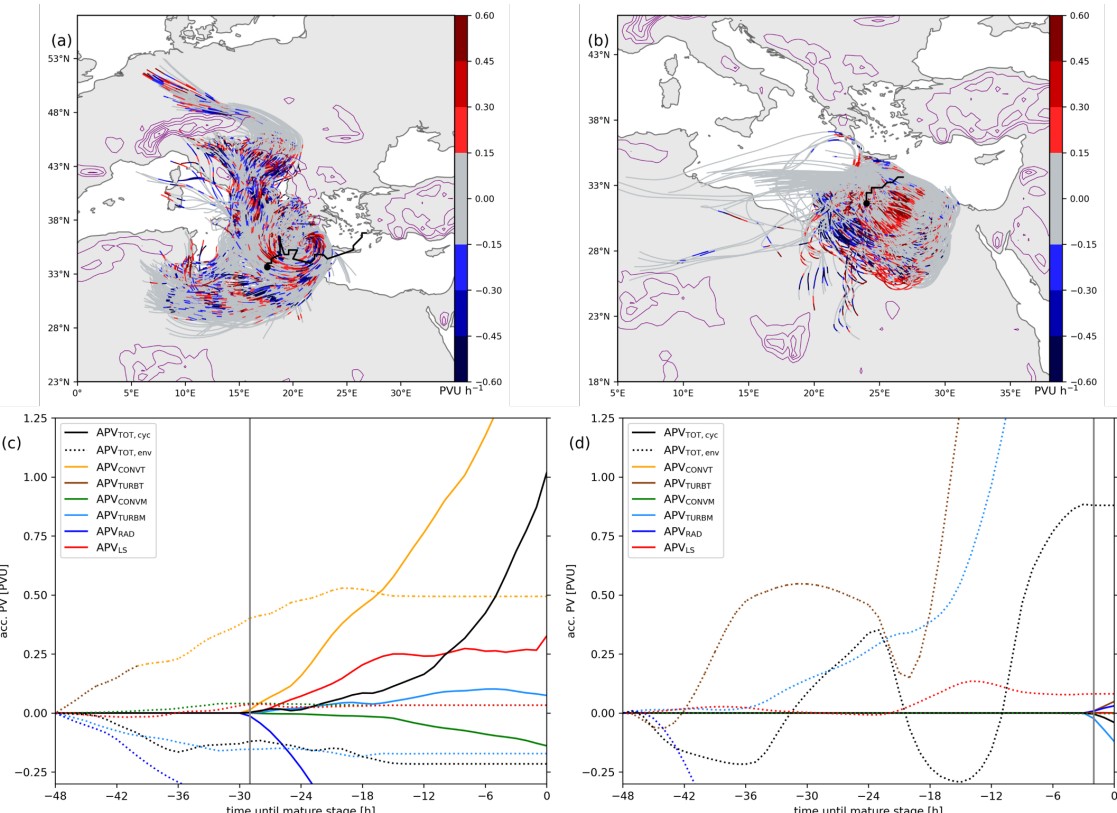

**Figure 11.** Hourly diabatic PV changes (colors) along trajectories initialized at the time of the cyclones's mature stage (black dot) at (a) 0200 UTC on 14 December 2017; (b) 0300 UTC on 19 June 2018 in the IFS simulations; (c,d) Time evolution of accumulated total PV (black) and accumulated PV of different processes (see legend) for the cyclones shown in (a,b). APV is further split into cyclonic and environmental components, represented by solid and dotted lines, respectively.

## 6   Diabatic processes in cyclonic and environmental PV in the IFS

Considering the 55 cyclones in the IFS simulations, we find comparable values of the average PV gain and the relative time interval of major PV production as for ERA5 cyclones. Furthermore, the average accumulated cyclonic and environmental PV modifications are approximately the same as in ERA5 and hence, although the set of cyclones in the IFS simulations is much
smaller, it represents a realistic sample of Mediterranean cyclones.

### 6.1   The contribution of diabatic processes to the lower-tropospheric PV

We present two contrasting cases of cyclones in the IFS simulations. Cyclone 1 (C1) had cyclogenesis over the central Mediterranean and reached its mature stage at 0200 UTC on 14 December 2017 near the south coast of Greece (black dot in Fig. 11a). It has three major trajectory groups: (i) starting over central Europe and bending around the Alps before passing over Italy, (ii)





starting over north Africa, and (iii) trajectories starting from over the Mediterranean Sea. On average, PV is mainly produced within the cyclone and the cyclonic PV (solid black line in Fig. 11c) accounts for 90 % of the PV anomaly (PV⋆), whereas the environment predominantly destroys PV (black dotted line in Fig. 11c). Furthermore, Fig. 11c shows the accumulated PV modifications due to individual diabatic processes inside the cyclone and in the environment (solid and dotted colored lines). The main driving process of PV production inside the cyclone is convection (solid orange line in Fig. 11c), which produces

2 PVU. The strong PV production by convection in the cyclone is reduced by radiative processes in the cyclone (solid dark blue line in Fig. 11c), which reduce PV by 1.25 PVU. The sum of all PV modifications inside the cyclone (solid colored lines) yield the total PV gain inside the cyclone of 1 PVU (black solid line in Fig. 11c). Radiative processes in the environment (dotted dark blue line in Fig. 11c) are the main source of PV destruction in the environment. Overall, the three distinct airstreams of C1 have similar contributions of diabatic PV modification but enter C1 at different times.

In cyclone 2 (C2), the trajectories start over Egypt and in general in proximity to the mature stage of C2 at 0300 UTC on 19 June 2018 (black dot in Fig. 11b). In contrast to C1, the lower-tropospheric PV in the cyclone center at the mature stage of C2 is mainly produced in the environment and accounts for approximately 60 % of PV⋆. This can be explained by the short time between cyclogenesis (vertical grey line in Fig. 11d) and the mature stage.

Additionally to the difference in cyclonic and environmental PV modifications, C2 has also two completely different main

driving processes. Figure 11d shows that the large PV changes in the environment (dotted black line) are entirely due to turbulence, i.e., turbulent momentum forcing (dotted light blue) and turbulent thermal forcing (dotted brown), with positive PV modifications of 1.75 and 1.6 PVU, respectively. Similarly to C1, radiation destroys PV in the environment (dotted blue line in Fig. 11d), with a negative contribution of −2.5 PVU, resulting in a net PV change by the environment of 0.85 PVU (dotted black line).

Generally, we find a large variability in the contribution of diabatic processes to the lower-tropospheric PV anomaly. Figure 12a shows the distribution of apv, the total and process-related APVs, and the residuals (RES) of the 55 cyclones in the IFS dataset. The integrated PV modification, APV$_{\text{TOT}}$, reproduces the real PV modification (apv) quite well, with some minor deviations in form of mainly negative residuals (RES). Therefore, the hourly PV tendencies provide a good representation of the occurring diabatic processes. On average, the largest PV production is due to the convective temperature tendencies (CONVT), turbulent

momentum tendencies (TURBM), and large scale microphysics (LS). In contrast, PV destruction is mainly due to turbulent temperature tendencies (TURBT) and radiation (RAD). Nevertheless, the boxes and whiskers reveal that there is a large case to case variability. These distributions are comparable in sign and strength to the PV modifications in extratropical cyclones in the main storm track region recently analyzed by Attinger et al. (2021). Furthermore, the overall strong positive contribution by latent heat release by both convection and large scale microphysics in the lower troposphere and the overall positive contribution

by turbulent momentum forcing with some individual negative cases is in accordance with findings by Flaounas et al. (2021). Splitting the overall contributions into the cyclonic and environmental parts (Fig. 12b,c), reveals that inside the cyclones (Fig. 12b) PV is mainly produced by convective temperature tendencies and microphysics. In the environment (Fig. 12c), convective temperature and turbulent momentum tendencies are the main processes producing PV.



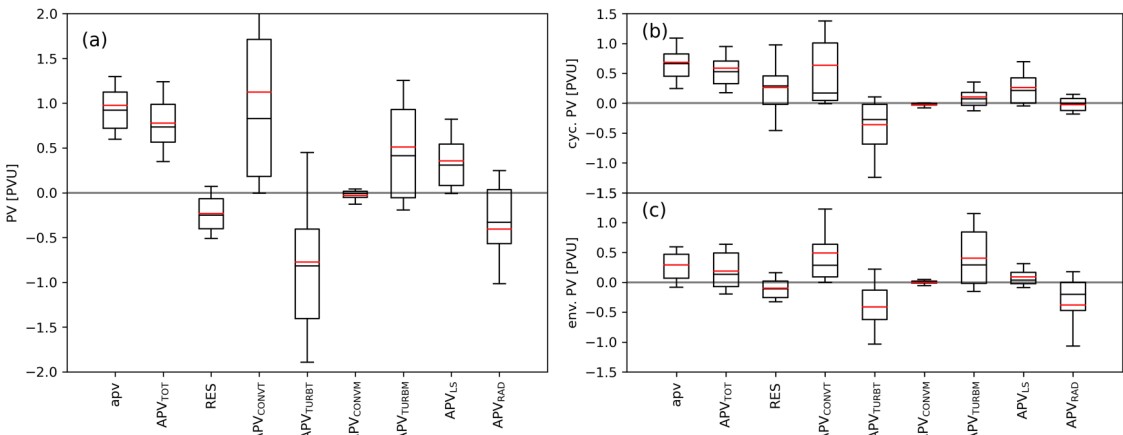

**Figure 12.** (a) Distribution of apv, residual and different APVs for the cyclones of the IFS simulations. (b,c) The distribution of (a) is split into its cyclonic (b) and environmental (c) part. The boxes mark the 25th and 75th, and the whiskers the $10^{th}$ and $90^{th}$ percentiles, the black and red lines inside the boxes depict the median and the mean, respectively.

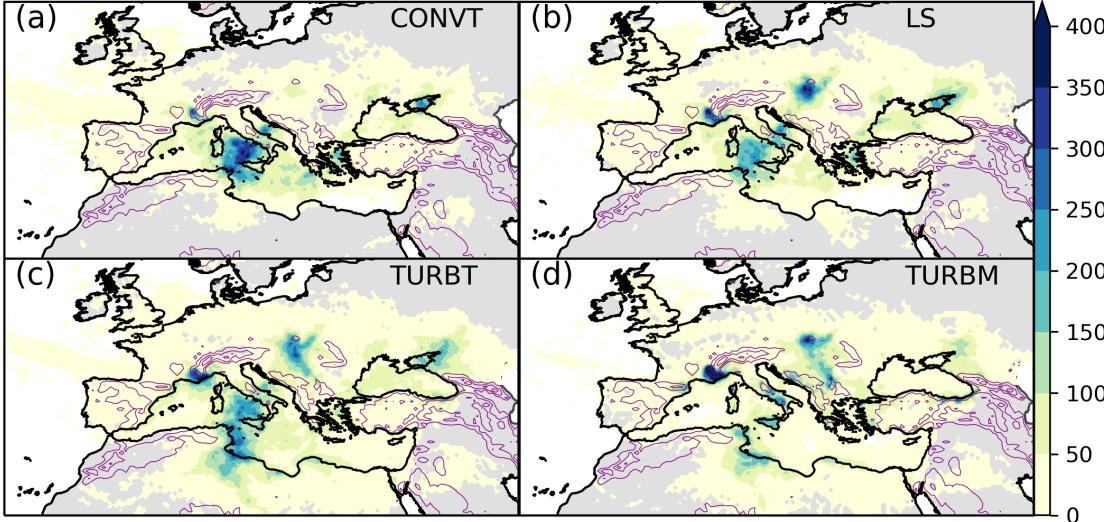

**Figure 13.** Absolute frequency of diabatic process related hourly PV change $\geq 0.15$ or $\leq -0.15\,\mathrm{PVU\,h^{-1}}$ from trajectories of IFS cyclones. Purple contours show elevation starting at $800\,\mathrm{m}$ with $800\,\mathrm{m}$ intervals.

With the trajectory data of the IFS cyclones, we cannot reproduce the climatological results of Fig. 7, due to fewer cyclones in the IFS. However, we can localize specific diabatic processes in the Mediterranean by showing regions with frequently large PV modifications ($\geq 0.15$ or $\leq -0.15\,\mathrm{PVU\,h^{-1}}$) due to different diabatic processes in Fig. 13.



Large PV modification due to latent heat release by convection and microphysics (Fig. 13a, b) predominantly occurs in the Tyrrhenian, Adriatic, Ionian, and Aegean Seas, and in the Black Sea, which are mainly positive changes. Additionally, microphysics led to large PV production over the European continent (Fig. 13b). Turbulence results in large PV changes near the
mountains, over Tunisia and Libya (Fig. 13c, d). For turbulent processes, the regions of frequent PV production and destruction tend to overlap, indicating that whether PV is produced or destroyed highly depends on the current local conditions. Systematic large hourly PV modifications by radiation are not frequent in our data of 55 IFS cyclones. However, over North Africa and the Caucasus mountains long-wave heating systematically destroys PV (not shown).

Finally, we identify regions that add to a large residual. Here, diabatic processes near the mountains, especially the Alps as
well as along coastlines provide the largest source of the residual (not shown). This coincides well with regions of large PV modifications by turbulent momentum forcing. This might be explained by the high variability of turbulent processes in time, and thus for turbulent processes, an hourly output might not be fully sufficient for our method, leading to an increased residual.

## 7    Discussion and conclusions

In this study we investigated where and when relative to the cyclone center and the mature stage of the cyclone the lower-
tropospheric PV anomaly in Mediterranean cyclones is produced. Therefore, we initialized 48 h backward trajectories from the lower-tropospheric PV anomaly in the cyclone center of 2969 mature cyclones in ERA5 and traced PV along them. Further, we introduced a new method to quantify, for the first time, cyclonic and environmental PV modifications.

We found that the lower-tropospheric PV anomaly is on average mainly produced by diabatic processes inside the cyclone center, within 12 h before the mature stage. It implies that the, for the intensity of the cyclone, essential production of lower-
tropospheric PV is a rather fast process. For most cyclones (59.2 %) the majority of the PV anomaly is produced inside the cyclone, which stresses the importance of the diabatic processes in cyclones, e.g. latent heat release by condensation, as highlighted by previous studies (e.g., Davis et al., 1993; Rossa et al., 2000; Čampa and Wernli, 2012). Nevertheless, we found a significant number of cyclones (37.3 %) for which the environment produces more than 50 % of the PV anomaly, stressing the importance of the processes in the environment for the development of those cyclones. Furthermore, for 300 cyclones
(10.4 %) the majority of the PV anomaly was traced back to environmental PV production near the mountains, which is the first systematic quantification of orographic PV production in Mediterranean cyclones. Case studies revealed the crucial role of mountains for the formation of specific cyclones, as almost the entire PV was produced near them. Therefore, the resolution of diabatic processes near mountains is crucial for correctly predicting the intensity of cyclones, as previously highlighted by Flaounas et al. (2022). Our climatological study confirmed that the lower-tropospheric PV anomaly can be produced to a large
degree in the environment.

The PV modifications near mountains might be related to so-called PV banners (Aebischer and Schär, 1998; Rotunno et al., 1999; Epifanio and Durran, 2002; Schär et al., 2003; Flamant et al., 2004). PV inside the PV-banners is produced by turbulence and diabatic processes within the boundary layer (Aebischer and Schär, 1998; Flamant et al., 2004) and can contribute to the lower-tropospheric PV in cyclones with cyclogenesis close to the Alps. Indeed Flaounas et al. (2021) could confirm this in case



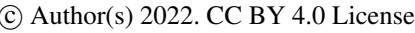

studies. We find this process to be present for up to $10\,\%$ of Mediterranean cyclones. It is interesting to note that many of these cyclones reached their mature stage shortly after cyclogenesis, which might indicate that these are rather weakly developing and/or short-lived systems. Therefore, adding an addtional criterion about the duration of intensification period when selecting the set of Mediterranean cyclones (e.g., requesting at least 12 hours between cyclogenesis and the mature stage), would, on average, reduce the importance of environmental (and orographic) PV production for the then selected cyclones. In other words,

orographic PV production is particularly important for cyclones that originate near mountains and then, after a typically short and weak intensification phase start to decay.

Our second focus was to quantify the contribution of individual diabatic processes to the PV production inside the cyclone and in the environment. Therefore, we analyzed 55 cyclones in twelve monthly IFS simulations. The approach is adequately accurate with a small average residual for individual cyclones and reveals that latent heat release by convection and micro-

physics produces most of the lower-tropospheric PV inside the cyclone, whereas in the environment turbulent momentum and convective temperature tendencies are the major sources of PV. However, the contribution by each process shows a very large case to case variability. Comparing with North Atlantic cyclones (Attinger et al., 2021), Mediterranean cyclones show a similar contribution of diabatic processes to their lower-tropospheric PV anomaly, which could indicate that Mediterranean cyclones do not form a unique category of extratropical cyclones, which is still an open question (Flaounas et al., 2022). However,

the cyclone sample in our study was much smaller than in Attinger et al. (2021) and thus might not be as representative. We provided a first hint on the preferred locations of sources and sinks of PV due to diabatic processes, which is the first time such an analysis has been performed for the Mediterranean.

There are some limitations concerning our study. First, our Lagrangian method of defining cyclonic and environmental PV heavily relies on the tracking of the cyclone. In case of rapidly developing systems, for which cyclogenesis might not be

resolved accurately, our assignment might be biased, attributing a larger amount of PV modifications to the environment rather than to the cyclone. Second, our cyclone sample in the IFS simulations is relatively small and thus the results have to be evaluated with caution. In addition, the hourly output of the IFS simulations might not be fully adequate for all cyclones, as diabatic processes, in particular turbulence, can have a very high temporal fluctuation and thus might not be captured accurately in the hourly outputs.

*Data availability.* The IFS simulations performed for this study can be provided by the authors upon request.

*Author contributions.* AS prepared all analyses and the manuscript. EF and HW provided scientific advice throughout the project and provided valuable suggestions for improving the manuscript.

*Competing interests.* Heini Wernli is executive editor of WCD.



*Acknowledgements.* AS acknowledges the funding from the Swiss National Science Foundation (Project 188660). Furthermore, we thank
Roman Attinger for help with the IFS data.





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
