# Peer review of "Origin of low-tropospheric potential vorticity in Mediterranean cyclones"

_Weather and Climate Dynamics, 2022_

## Referee Comment (RC2)

**General comments**:

The paper seeks to investigate how PV is generated in Mediterranean tropical cyclones 'medicanes' by using a Lagrangian backward trajectories from a defined mature state over the previous 48 hours for the IFS model and ERA5. The paper describes the relative contributions of accumulated PV both from the environment of the cyclone, the cyclone itself and any initial PV anomaly. The paper then further tries to isolate the contributions of both environmental and cyclonic PV by breaking it down into diabatic components such as convection and microphysics. The authors also spend some time talking about PV generated from orography and its contribution to environmentally generated or destroyed PV. The authors do a good job of generalizing their work by including composite plots of their key results and relevant representative case studies.

Overall the paper presents a fairly compelling story that for most developing medicanes, the principle generation of PV is from diabatic generation inside the cyclone itself, and from the convection in particular. The authors are careful to note, that a significant minority of storms do generate much of their PV from the environment including orographic production of PV.

This study does provide useful novel results on the generation of PV in medicanes, which are understudied storms in general. I think the paper could be stronger, however, if the authors justified why the generation of PV is something we should care about so these new results can be contextualized.

I do have some concerns with the methodology notably with the "cyclone effective area" parameter being considerably larger than the eyewall radius of a typical medicane. As such I think there needs to be further discussion about how the grid spacing of the models could affect the results, and how some of the cyclonically generated PV likely comes from thunderstorms outside of the medicane core rather than the small ring of convection near the centre.

Another related issue I have is that the authors do not attempt to account for the changing structure of the storms throughout their development. Unlike a normal tropical cyclone which may have a deep warm core from cyclogenesis to landfall a medicane inevitably transitions from an extratropical deep cold core low, through to a large subtropical low with scattered thunderstorms, to finally a tiny, tightly bound tropical structure with a self-sustaining shallow warm core. It should not be assumed PV generation throughout these different life stages are the same, and I suspect they could be very different. A way to address this concern would be some stricter filtering that only contains storms that have a tropical phase (some will only ever be subtropical) and then identify the times at which this transition occurs. You should try to ensure that your definition of 'maturity' occurs after the storm has entered its tropical phase. There are some other concerns I have with this definition to which I have outlined in the detailed comments. Other than these concerns my only other substantive issue is that there is sometimes a lack of clarity in the specific PV related quantity being referred to. I'd suggest being more explicit about things like whether the PV has been averaged over all of the back trajectories, or that you are referring to diagnosed PV and so on.

I think this is an interesting paper with robust use of the back trajectory methodology applied to an underappreciated context. I would recommend this paper be accepted once the concerns have been

addressed.

**Specific comments**

- L57 has a 'mature' medicane been defined? I think this is important, everyone knows what a mature tropical cyclone is, but medicanes have a very long transition (relative to their lifetime) from extratropical to subtropical to tropical like. So I think 'mature' needs to be more clearly defined. You first mention maturity in L29 where you say *"Thereby, the typically (very) strong cyclonic circulation in the lower troposphere in the mature stage of extratropical and Mediterranean cyclones can be explained by the distinct vertical alignment of the diabatically produced positive PV anomaly at low levels with the upper-level PV streamer or cutoff, forming a so-called PV tower"* so you have referred to a characteristic of 'mature' medicanes, but the word 'mature' is not explicitly defined prior. You did define it later at L90 however it feels like it is a bit late.

- L77. Medicanes are very small (in their tropical phase), 0.4 degrees would have the entire inner core on possibly a single grid point. This may be a major limitation of the study, since everything might be the 'environment'.

- L94. You don't mention a means of filtering out purely extratropical cyclones from medicanes. I suspect there are some extratropical cyclones in your study (or ones that are pre tropical or post tropical when 'mature') as there are rather a large number of red dots over land. Medicanes decay very quickly over land (even faster than a normal TC), though they may then intensify as post tropical cyclones with frontal systems so I doubt the red dots over Tunisia still have tropical characteristics by the time they have moved so far inland and, as a fundamentally different type of storm, their dynamical processes will also be different. You may also get an issue where the storm is strongest (and 'mature' by your definition) in its pre-tropical phase. I wouldn't be surprised if a medicane precursor over the Linguarian sea has an initially much stronger relative vorticity as a 970mb extratropical frontal cyclone than 48 hours later as a 995mb tropical like medicane. You may be able to filter these cases out manually but I think it needs checking and mentioning in your methodology from your figure 2b I'd be the most wary of any storms that are below 990mb which is strong for a medicane.

- L101. A quick note bene which you may already have considered. A medicane can actually increase its MSLP but still be intensifying because the 'upper trough envelope' is decaying faster than the medicane is intensifying, so the pressure gradient near the medicane centre can go up and the absolute value of the MSLP can also go up. This is an intricacy that might mean your T+0h is actually before many of your medicanes are at their strongest. I don't think there is anything wrong with how you define T+0h (apart from my previous comment) but the reader should be aware that it doesn't necessarily correspond to when the medicane is at its strongest.

- L119. One way you could extend this work (in, perhaps, a future paper) is to use MetUM simulations of chosen medicanes which have built in PV lagrangian tracers in addition to a higher spatial resolution.

- L140. Since you are initiating your backward trajectories using your prior T+0 definition, some further consideration of my comment for L101 might be useful since your results might be stronger if T+0 did, indeed, correspond to the medicane at its strongest. You could, include wind speed or tangential wind speed in your definition of maturity for example or use a radial gradient of SLP rather than the MSLP.

- L151. You do now do this (previous comment), I think it would be easier for the reader if this came slightly earlier. These first couple of sentences do indeed help my understanding of what you are doing a lot.

- L172. I think you have a reasonable justification for your "cyclone effective area" parameter here, but I am still a little nervous about how large it is. 400km might be reasonable in a normal Atlantic or Pacific TC (although even then it feels on the large side) but medicanes are very small, which is why Hart phase space diagrams struggle to denote them as tropical with the usual calculation domain size. The two papers you reference in the justification both also use fairly low resolution spatial data (1.125 degrees). In model simulations and observations the 'eyewall' and RMW is much smaller than this (most likely much less than 0.5 degrees)  so will the radius of any PV modification from the diabatic processes associated with this convection be. I think, as a result, of how these convective processes are parameterized in this lower resolution data, you would expect the effective core size to appear larger than it is in a real medicane. Nevertheless, I feel like this needs to be explicitly acknowledged here and in a limitations section, particularly since a future (higher resolution) modelling or observational study might find very different and considerably smaller 'cyclone effective area' sizes if an analogous method, based on this work, is chosen to be adopted by another researcher.

- Have you checked diurnal variation in Radiative PV changes (particularly cyclonic), we know medicanes are much stronger at night, so I wonder if this is also visible in the dark blue line (which we cannot see because it is cropped). I feel like it is possible there may be a very interesting result hidden here. Even if you don't spot a diurnal cycle since this has been observed before it is definitely worth a sentence on.

**Technical corrections:**

- L150. Could be clearer about how many of these backward trajectories are going to be initialized in your study. In your example you initialize 3 but I'm assuming this number is not special, I think it would be clearer to more explicitly say that Fig 3 uses three backward trajectories so it doesn't feel like the number 3 is integral to the method.

- L162. Could be really nitpicking here, but would it read better if you swapped the order of (i) and (ii) since (ii) is this backward trajectory method that you have just spent the last section and the first part of this section explaining in detail whilst (i) is ancillary and doesn't need any explanation. Also since you haven't talked about (iii) yet the grammar seems slightly weird. 'For (iii)' makes it seem like you are referring to something hitherto mentioned. Perhaps 'In addition we also pragmatically define (iii), a "cyclone effective area"' reads better.

- L178. you are defining 'apve' and 'apvc' now but they also appear in your Figure 3 when they are not defined. I would expand your Figure 3 caption to deal with this. Additionally, I'd be more clear about the coloured lines being the *diagnosed* PV and the other lines being cumulative components integrated over time. Another issue with Figure 3 is the scale goes off the bottom in (c ).

- L187: how do you know the cyclone both 'produces' and 'destroys' PV resulting in the net negative cyclonic PV? The cyclone could entirely destroy PV (albeit intuitively unlikely – is this what you mean) and still cause the same result to be seen, is there unseen work that led you to this conclusion?

- L194 the word 'remaining' makes me think of budget residuals, while I believe the intention is to refer to the positive environmental PV left over after the large peak at around T-45h (so 'remaining' here refers to a change in time of one component rather than some difference involving multiple components). I'd possibly think of a way of rephrasing to make it clearer. It is rather unfortunate that the difference between the top black line and the solid grey line is also coincidentally around 0.5PVU, be aware a reader might, incorrectly, think this is what you are referring to.

- L196 *"It is 0.5 PVU of orographic PV and not 1.5 PVU, because the environmental PV at t = −38 h, which is completely orographic (dotted and grey lines), is reduced by the environment between t = −38 h and t = −22 h in absence of orography."*. I understand what you are trying to say but am not sure its semantically true. The orographic PV surely *is* 1.5PVU but also combines with a -1 non orographic PV (which you say in L198) to give the overall 0.5 environmental PVU at T+0h. It is true that this positive 0.5PVU remnant is entirely orographically generated as you have shown, but calling it the 'orographic PV' feels misleading. I wonder if the overall points might be clearer if, instead, of partitioning your components into 'orographic', 'environmental' and 'cyclonic' you have 'environmental orgographic', 'environmental non orographic' and 'cyclonic'. That way the components intuitively add up to give the diagnosed PV and the figures are easier to read and understand on the first reading. I suspect your explanation will also be shorter and clearer.

- L205 this is the first time you mention the 'core' of the medicane. Is this the same as your 'cyclone effective area' for T=0h? Also do you initialize a back trajectory at every gridpoint in this region? If you do, it isn't clear.

- Fig 5b could be clearer, I don't think, for example, you explicitly refer to your solid grey PV line (I think it needs to be explicitly said that this is an average of (a) the **diagnosed** PV over (b) **the trajectories**).

- L295: "*After the trajectories enter the cyclone effective radius (vertical grey line)*". I assume this vertical grey line is the **average** time the trajectories enter the cyclone effective radius (as they will presumably happen at slightly different times).

- L300: Your 2$^{nd}$ case study involves a cyclone reaching maturity in the Black sea. I have no problem with this, the tropical-like storms that occasionally form in the black sea are structurally similar to medicanes however you don't mention the black sea in your introduction. It might be worth including a sentence so the reader is not surprised by the location of this case study.

- L330: The yellow line goes off the scale so I can't see that it produces 2PVU as stated. Also, to be picky, the line looks more yellow than orange. The blue line also goes off the scale. I understand, as a result, we get a zoomed in version of the other lines, but I'm not sure it can justify cropping out this data.

- L346. A little unclear about the difference between small case apv and APVtot. I'm assuming apv is simply the difference between the diagnosed PV at T+0h and T-48h?

---

## Author Comment (AC1)

wcd-2022-40

**Origin of low-tropospheric potential vorticity in Mediterranean cyclones**

by Alexander Scherrmann, Heini Wernli, and Emmanouil Flaounas

**Final author comments**

We would like to thank both Reviewers for their constructive comments, suggestions and remarks that helped us to improve the manuscript. Below are our detailed replies (in blue) to the individual comments (in black).

We would like to thank both Reviewers for carefully reading our manuscript and the fruitful comments they provided. Most comments have been incorporated including the following major change:

We now split the environmental PV as suggested by Reviewer 2 into a non-orographic environmental and orographic environmental part. This allowed a more precise identification of orographic cyclones which increased the number of so-called orographic cyclones from 300 to 580. Their climatology remains similar to the previously identified one. The non-orographic and orographic PV almost contribute evenly to the total environmental PV. However, the orographic environmental PV is mainly a large contribution in distinct airstreams and not present in most other trajectories.

**REVIEW 1**

Line 15 and elsewhere: "large-scale microphysics": since microphysics occurs at small scales, the expression appears rather contradictory: please use a different term to represent these processes;

Changed it to "microphysics".

Line 32: About PV tower in the Mediterranean, please consider also Miglietta et al. (2017):

Miglietta M.M., D. Cerrai, S. Laviola, E. Cattani, V. Levizzani, Potential vorticity patterns in Mediterranean "hurricanes", Geophysical Research Letters, 44, 2537-2545, 2017, http://dx.doi.org/10.1002/2017GL072670;

Thank you for the additional reference, included as suggested.

Line 75: some additional indications should be provided about the way the monthly simulations are forced; for example, how is the deviation of the numerical simulations from real conditions prevented?

The "real atmospheric state" is only prescribed in the initial conditions. Being global simulations, lateral boundary conditions cannot be applied and no nudging has been used

to relax the model output to the reanalysis. Therefore, the model deviates from real conditions after the first 3-6 days. Nevertheless, the model still produces physically realistic systems and atmospheric processes and thus the dynamical analysis of the simulated cyclones is still valid (see also Attinger et al., 2021). We are not comparing these cyclones in the simulations to their counterparts in the real world, but we rather analyze their dynamics. We clarify this now in the text.

Line 93: I guess "large" stands here for wide horizontal extent, not reaching high values, right? Please explain better.

Thank you for the remark. It is indeed a combination of amplitude and horizontal extent. We now better explain this as follows:
"Finally, we select cyclones where low-level PV reaches both a certain spatial size and amplitude, such that we can initialize at least 200 backward trajectories at grid points with PV $> 0.75$ PVU from their mature stage."

Line 144: The choice of the threshold of 0.75 PVU appears rather arbitrary: did you check with other thresholds, for example did you try how would the results change using 1 PVU?

Increasing the threshold to 1PVU, i.e., to nearly three times the climatological background value, significantly reduces the number of cyclones for which we can initialize 200 or more trajectories from 2969 to 1809. For those cyclones, the results would change accordingly:

1. The climatology remains fairly similar, and the cyclones have similar intensities and life times.
2. The average PV evolution is almost identical when choosing a threshold of 0.75 PVU or 1PVU, and so it is for the cyclonically and environmentally produced PV. The figure below is the same as Fig. 6 in the paper, however for the 1 PVU threshold

[Figure]

Therefore, increasing the threshold provides similar results but reduces the sample of the analyzed systems. Furthermore, the composite analysis by Flaounas et al. (2021; their Fig.

5d) shows PV values at 850 hPa at the mature stage of about 0.7-0.8 PVU.  We now refer to their results to better motivate our threshold of 0.75 PVU.

Flaounas, E., Gray, S. L., and Teubler, F.: A process-based anatomy of Mediterranean cyclones: from baroclinic lows to tropical-like systems, Weather Clim. Dynam., 2, 255–279, https://doi.org/10.5194/wcd-2-255-2021, 2021

Line 196: please remove "of orographic PV", I think it is confusing here;

Changed as suggested

Line 199: I think "environmental PV" is more appropriate than "orographic PV" here;

We are now more explicit in our analysis by including the concept of non-orographic environmental PV, as suggested by Reviewer 2. For details please see our reply to the comment from Reviewer 2 about non-orographic environmental PV – L196 "It is 0.5 PVU of orographic…" below.

The environmental PV contributes at the time of the mature stage with 0.5 PVU. Splitting the environmental PV results in a non-orographic component of −1 PVU and an orographic component of +1.5 PVU, respectively. Therefore, we consider the +0.5 PVU total environmental contribution as mainly related to the orographic component and thus refer to it as orographic environmental PV.

Line 250-251: "By design the ratio of the cyclonic and environmental contributions is the same when considering the total lower-tropospheric PV or the PV anomaly": please clarify this sentence.

Thank you for the remark, the text now rereads:
"The cyclonic and environmental contribution to the total lower-tropospheric PV anomaly is 40 % and 23%, respectively. Accordingly, the contributions to the PV anomaly are 56 % and 32%. The ratios 40/23 and 56/32 are nearly identical by design."

Line 254: please change into "have **at the same time** 50%".

Changed as suggested.

Line 265: … **are** less frequent …

Added as suggested

Line 279-281: Another hot spot region is central-eastern Alps and the northern Adriatic Sea: some comments on that area?

We will explicitly mention these regions in the revised manuscript.

Line 297: the remaining 25% **of anomaly is** …

    Added as suggested

Figure 9 caption: from the text the mature stage is at 0400 UTC on 13 December 1988, not at 1300 UTC on 12 December 1988.

    Thank you for the remark, changed as suggested.

Line 359-360: Since it is not possible to reproduce the results of Fig. 7, why should one trust the results of Fig. 13? I think this part should be removed or better motivated.

    We agree that line 359 is not properly reflecting our motivation.

    Due to their limited duration, the simulations performed with the IFS are not adequate to perform a climatological analysis and therefore we cannot produce a figure like Fig. 7 with the IFS simulations. However, the IFS simulations offer additional information about the tendencies of physical parameterizations. Therefore, we use the IFS simulations as a complementary dataset to gain further insights into the areas that favor the non-conserved PV processes. We now reformulated the text accordingly.

Line 371-372: please explain better this point;

    We removed lines 369-372 as they seem to confuse the reader and are not essential for the results.

Line 395: present or dominant?

    We rephrased as follows:

    "We find this process to significantly contribute by at least 25 % to the lower-tropospheric PV anomaly in 19.5 % of the Mediterranean cyclones, i.e. in 580 cyclones from a total of 2969."

    The numbers changed according to the analysis performed in response to a comment from Reviewer 2, please see below our reply to the comment about non-orographic environmental PV – L196 "It is 0.5 PVU of orographic…".

**REVIEW 2**

Several of the comments of reviewer 2 refer to medicane systems, i.e. tropical-like cyclones that develop in the Mediterranean. However, our study has no special focus on these systems. We constantly refer to "Mediterranean cyclones" and only once or twice in the entire manuscript to

"medicanes" or "tropical-like cyclones". In fact, medicane systems are still undefined in term of physical criteria (see discussion in the review paper by Flaounas et al., 2022). With an empirical annual frequency of 1-3 medicanes per year, they only represent a very minor fraction of the cyclones that we analyze here. Nevertheless, we consider that the majority of the comments stand true for all Mediterranean cyclones analyzed in our study. With this assumption we reply to the queries below.

L57 has a 'mature' medicane been defined? I think this is important, everyone knows what a mature tropical cyclone is, but medicanes have a very long transition (relative to their lifetime) from extratropical to subtropical to tropical like. So I think 'mature' needs to be more clearly defined. You first mention maturity in L29 where you say "Thereby, the typically (very) strong cyclonic circulation in the lower troposphere in the mature stage of extratropical and Mediterranean cyclones can be explained by the distinct vertical alignment of the diabatically produced positive PV anomaly at low levels with the upper-level PV streamer or cutoff, forming a so-called PV tower" so you have referred to a characteristic of 'mature' medicanes, but the word 'mature' is not explicitly defined prior. You did define it later at L90 however it feels like it is a bit late.

> We now mention in lines 29 and 57 that maturity refers to the time of maximum intensity. To avoid confusion since not all intense cyclones in the Mediterranean may produce prominent PV anomalies of diabatic origin in the lower troposphere, we slightly rephrased our formulation to:
> "Thereby, the typically strong cyclonic circulation in the lower troposphere in the mature stage of extratropical and Mediterranean cyclones can be often explained by the distinct vertical alignment of the diabatically-produced positive PV anomaly at low levels with the upper-level PV streamer or cutoff, forming a so-called PV tower".

L77. Medicanes are very small (in their tropical phase), 0.4 degrees would have the entire inner core on possibly a single grid point. This may be a major limitation of the study, since everything might be the 'environment'.

> As explained before, medicanes are still physically undefined and in most known cases of such systems there is a baroclinic forcing that is far from being negligible (see Flaounas et al., 2021; 2022). Nevertheless, even if we consider that medicanes are "purely" developed through convective processes, similarly to their tropical counterparts, then still a model with a grid spacing of 0.4 degrees would capture this diabatic development through physical parameterization (as for instance is the case in most reanalyses). Therefore, interpretation in terms of processes would be fairly similar to a model where convection is explicitly resolved, albeit less detailed. In our case, the environmental PV production is considered to take place beyond the area defined by a radius of 400 km around the cyclones center. As a result, even if there was a convective eyewall, the associated PV production would still be considered as "cyclonic".

L94. You don't mention a means of filtering out purely extratropical cyclones from medicanes. I suspect there are some extratropical cyclones in your study (or ones that are pre tropical or post tropical when 'mature') as there are rather a large number of red dots over land. Medicanes decay very quickly over land (even faster than a normal TC), though they may then intensify as post tropical cyclones with frontal systems so I doubt the red dots over Tunisia still have tropical characteristics by the time they have moved so far inland and, as a fundamentally different type of storm, their dynamical processes will also be different. You may also get an issue where the storm is strongest (and 'mature' by your definition) in its pre-tropical phase. I wouldn't be surprised if a medicane precursor over the Linguarian sea has an initially much stronger relative vorticity as a 970mb extratropical frontal cyclone than 48 hours later as a 995mb tropical like medicane. You may be able to filter these cases out manually but I think it needs checking and mentioning in your methodology from your figure 2b I'd be the most wary of any storms that are below 990mb which is strong for a medicane.

> We consider this comment to be based on a misunderstanding. As explained in the beginning of our reply to the second Reviewer, we do not focus explicitly on medicanes, and therefore there is no need to separate medicanes from what the reviewer refers to as extratropical cyclones. We simply consider all intense Mediterranean cyclones.

L101. A quick note bene which you may already have considered. A medicane can actually increase its MSLP but still be intensifying because the 'upper trough envelope' is decaying faster than the medicane is intensifying, so the pressure gradient near the medicane centre can go up and the absolute value of the MSLP can also go up. This is an intricacy that might mean your T+0h is actually before many of your medicanes are at their strongest. I don't think there is anything wrong with how you define T+0h (apart from my previous comment) but the reader should be aware that it doesn't necessarily correspond to when the medicane is at its strongest.

> It is not clear to us what is meant by "decay of an upper trough envelope".

> If we consider a simplified scheme where the cyclonic circulation of a low pressure system such as a medicane is the outcome of two forcings: a baroclinic one (for instance from a through) and a diabatic one (from convection in the center of the cyclone), then we consider that the Reviewer suggests that the "*MSLP at the cyclone center would increase because of a decrease of baroclinic forcing. However, the MSLP gradient close to the cyclone center and thus the wind speed would still intensify due to diabatic forcing*". By "intensify" we consider that the Reviewer is indeed referring to maximum wind speed close to the center of cyclonesand by "pressure gradients" we also take it as a given that the Reviewer considers medicanes to develop distinctive convective eyewalls and consequent symmetric "doughnut-like" wind speed structures.

> With these considerations we remark the following:
> The scales affected by baroclinic forcing from an upper-tropospheric system are typically larger than the ones affected by convection. Also, it is rather delicate to consider that the MSLP anomaly imposed by baroclinic forcing always collocates with the minimum MSLP at the cyclone center. Therefore, a "decay of an upper trough envelop" may affect the pressure gradients and central minimum MSLP in many unsystematic ways. As a result, it would be very rare for a Mediterranean cyclone to comply with the mechanism

described by the Reviewer. Still though such a mechanism would not apply to our study since a supposing convective eyewall would be almost falling to the category of subgrid scale structures -as previously stressed by the Reviewer- given the 0.4 degree resolution.

L140. Since you are initiating your backward trajectories using your prior T+0 definition, some further consideration of my comment for L101 might be useful since your results might be stronger if T+0 did, indeed, correspond to the medicane at its strongest. You could, include wind speed or tangential wind speed in your definition of maturity for example or use a radial gradient of SLP rather than the MSLP.

Please refer to our reply in the previous comment.

L172. I think you have a reasonable justification for your "cyclone effective area" parameter here, but I am still a little nervous about how large it is. 400km might be reasonable in a normal Atlantic or Pacific TC (although even then it feels on the large side) but medicanes are very small, which is why Hart phase space diagrams struggle to denote them as tropical with the usual calculation domain size. The two papers you reference in the justification both also use fairly low resolution spatial data (1.125 degrees). In model simulations and observations the 'eyewall' and RMW is much smaller than this (most likely much less than 0.5 degrees) so will the radius of any PV modification from the diabatic processes associated with this convection be. I think, as a result, of how these convective processes are parameterized in this lower resolution data, you would expect the effective core size to appear larger than it is in a real medicane. Nevertheless, I feel like this needs to be explicitly acknowledged here and in a limitations section, particularly since a future (higher resolution) modelling or observational study might find very different and considerably smaller 'cyclone effective area' sizes if an analogous method, based on this work, is chosen to be adopted by another researcher.

These considerations are all valuable and well thought when studying medicanes or any other small-scale weather system. However, our study focuses on all Mediterranean cyclones and, Medicanes represent only a minor fraction.

As a general remark, we would like to stress that Mediterranean cyclones are mesoscale systems and therefore a 400 km radius may be also considered rather small. We invite the Reviewer to consider structural scales of the order of 400 km in composite structures of intense Mediterranean cyclones such as the ones shown in Flaounas et al. (2015; their Fig. 3) but also in composite structures of medicanes such as the ones in Zhang et al. (2021). Especially in the latter study, medicanes have been selected using pressure gradient thresholds and Hart phase space diagrams. Despite the use of criteria that are adequate for tropical cyclones, composite structures in their Fig. 2 show indeed a baroclinic environment and precipitation patterns (therefor also diabatic processes) that extend beyond a presumed small area of a supposing convective eyewall. As a result, even Mediterranean cyclones with matching criteria traditionally used to distinguish tropical from extratropical cyclones may still be related to diabatic processes in an extended area with a 400 km radius.

Flaounas, E., Raveh-Rubin, S., Wernli, H. *et al.* The dynamical structure of intense Mediterranean cyclones. *Clim Dyn* **44**, 2411–2427 (2015). https://doi.org/10.1007/s00382-014-2330-2

Zhang, W, Villarini, G, Scoccimarro, E, Napolitano, F. Examining the precipitation associated with medicanes in the high-resolution ERA-5 reanalysis data. *Int J Climatol*. 2021; 41 (Suppl. 1): E126– E132. https://doi.org/10.1002/joc.6669

L119. One way you could extend this work (in, perhaps, a future paper) is to use MetUM simulations of chosen medicanes which have built in PV lagrangian tracers in addition to a higher spatial resolution.

Thank you for the suggestion.

L151. You do now do this (previous comment L140), I think it would be easier for the reader if this came slightly earlier. These first couple of sentences do indeed help my understanding of what you are doing a lot.

We now mention in the beginning of Section 3 the definition of cyclonic and environmental PV changes.

Have you checked diurnal variation in radiative PV changes (particularly cyclonic), we know medicanes are much stronger at night, so I wonder if this is also visible in the dark blue line (which we cannot see because it is cropped). I feel like it is possible there may be a very interesting result hidden here. Even if you don't spot a diurnal cycle since this has been observed before it is definitely worth a sentence on.

This is an interesting comment. We addressed the question of a diurnal cycle of radiative PV modification by adjusting the scale of Fig. 11c (see below) and for all IFS cyclones, but we did not find such evidence (dark blue line).

[Figure]

Technical corrections:

L150. Could be clearer about how many of these backward trajectories are going to be initialized in your study. In your example you initialize 3 but I'm assuming this number is not special, I think it would be clearer to more explicitly say that Fig 3 uses three backward trajectories so it doesn't feel like the number 3 is integral to the method.

> In this section we just provide an example set of three trajectories to explicitly explain the methodology. However, mentioning the average number of backward trajectories per cyclone is a valid suggestion. On average there are 590 initialized trajectories per cyclone. We now mention this in the text.

L162. Could be really nitpicking here, but would it read better if you swapped the order of (i) and (ii) since (ii) is this backward trajectory method that you have just spent the last section and the first part of this section explaining in detail whilst (i) is ancillary and doesn't need any explanation. Also since you haven't talked about (iii) yet the grammar seems slightly weird. 'For (iii)' makes it seem like you are referring to something hitherto mentioned. Perhaps 'In addition we also pragmatically define (iii), a "cyclone effective area"' reads better.

> Changed as suggested.

L178. you are defining 'apve' and 'apvc' now but they also appear in your Figure 3 when they are not defined. I would expand your Figure 3 caption to deal with this. Additionally, I'd be more clear about the coloured lines being the diagnosed PV and the other lines being cumulative components integrated over time. Another issue with Figure 3 is the scale goes off the bottom in (c ).

Thank you for this remark. We now refer to their definition in the caption and adjusted the y-axis of Fig. 3c (see below). Please also see the suggested splitting between non-orographic (dotted black line) and orographic environmental PV (dotted grey line) referring to your suggestion in a comment below.

[Figure]

L187: how do you know the cyclone both 'produces' and 'destroys' PV resulting in the net negative cyclonic PV? The cyclone could entirely destroy PV (albeit intuitively unlikely – is this what you mean) and still cause the same result to be seen, is there unseen work that led you to this conclusion?

For the single trajectory (Fig. 3c, see above), the changes in PV (blue) after t = −27 h almost entirely occur within the cyclone effective area (exception at t = −22 h). Therefore, any of the following PV changes are viewed as caused by the cyclone (except at t = −22 h). The decline in PV between t = −27 h and t = −21 h is interpreted as PV destruction by the cyclone (also shown by very negative apvc in this period (out of scale), solid black line), whereas the PV increase between t = −20 h and t = −6 h is viewed as PV production by the cyclone (increase in apvc, solid black goes up to ~0 PVU again). Between t = −6 h and t = 0 h PV is again destroyed by the cyclone, leading to a negative apvc at t = 0 h. Therefore, summing all PV changes (production and destruction) by the cyclone (apvc) yields an overall/net PV destruction by the cyclone. We now further clarify this in the manuscript.

L194 the word 'remaining' makes me think of budget residuals, while I believe the intention is to refer to the positive environmental PV left over after the large peak at around T-45h (so 'remaining' here refers to a change in time of one component rather than some difference

involving multiple components). I'd possibly think of a way of rephrasing to make it clearer. It is rather unfortunate that the difference between the top black line and the solid grey line is also coincidentally around 0.5PVU, be aware a reader might, incorrectly, think this is what you are referring to.

> Thank you for this comment. Also in response to a comment from Reviewer 1, we adapted the text starting from L193:
>
> "To give an example, the trajectory in Fig. 3d experiences a steep early increase of environmental PV when it is located over the Dinaric Alps (purple contours in Fig. 3a). In this case, the early PV increase (after t = −48 h) is related to orographic PV modification (grey line) and thus the 1.5 PVU environmental PV at t = −38 h entirely consists of orographic PV. Afterwards, there are no further orographic PV changes (horizontal grey line) and the environmental PV is modified by non-orographic PV changes, resulting in a final environmental PV of 0.5 PVU at t = 0 h. As the apve always remains larger than 0 PVU, i.e. the orographic PV is never completely destroyed, we thus refer to the final 0.5 PVU environmental PV as orographic PV."

L196 "It is 0.5 PVU of orographic PV and not 1.5 PVU, because the environmental PV at t = −38 h, which is completely orographic (dotted and grey lines), is reduced by the environment between t = −38 h and t = −22 h in absence of orography.". I understand what you are trying to say but am not sure its semantically true. The orographic PV surely is 1.5PVU but also combines with a -1 non orographic PV (which you say in L198) to give the overall 0.5 environmental PVU at T+0h. It is true that this positive 0.5PVU remnant is entirely orographically generated as you have shown, but calling it the 'orographic PV' feels misleading. I wonder if the overall points might be clearer if, instead, of partitioning your components into 'orographic', 'environmental' and 'cyclonic' you have 'environmental orographic', 'environmental non orographic' and 'cyclonic'. That way the components intuitively add up to give the diagnosed PV and the figures are easier to read and understand on the first reading. I suspect your explanation will also be shorter and clearer.

> Thank you very much. This is an excellent suggestion. We redid the climatological analysis of the ERA5 cyclones with the suggested splitting between "orographic environmental" (hereafter apveO) and "non-orographic environmental" (apveNO) PV and came to the following conclusions:
>
> 1. With the additional apveNO we could define the orographically influenced cyclones more precisely, which significantly increased the number from about 300 to 580 cyclones to which the apveO provides a minimum of 25 % of the PV anomaly. The climatological location of these cyclones remains very similar (see figure below and compare with Fig. 8 in the original submission).

[Figure]

2. Splitting apve in its apveNO and apveO parts leads to the following version of Fig. 6b.

[Figure]

The red lines are the same as in the original figure. The blue and grey lines and shadings now represent the non-orographic apve (apveNO) and orographic apve (apveO), respectively. The percentiles (shading and dashed blue lines) of the apveNO are very similar to the original apve, especially at early times. However, the mean value of apveNO (solid blue) at t = 0 h reaches only half of the previous value with 0.2 PVU and the 10[th] percentile leaves the original scale indicating a very large non-orographic related PV destruction, which was compensated by orographic PV production in the non-split scenario. A slightly lower but similar mean value is provided by the apveO at t = 0 h (solid grey). However, it results from fewer but therefore stronger orographically influenced trajectories, as more than half of the trajectories are not affected by orography

at all (25th and 75th percentiles (dashed grey line) have a PV value of 0 PVU). This shows that an interaction of airstreams and orography occurs in distinct airstreams rather than in all trajectories. Since this version of Fig. 6b is rather busy, we decided to keep the original version of Fig. 6b in the paper (with total apve only), but we discuss the split between apveO and apveNO in the revised manuscript.

Considering the 500 orographically influenced cyclones, we find a very steep cyclonic PV production of 0.3 PVU shortly prior to the mature stage, indicating that the trajectories enter the cyclone rather late. The grey shading and dashed line (75th percentile) show that the orography can provide high-valued positive PV to the lower-troposphere in these cyclones. Nevertheless, the 25th percentile and the late increase in the 75th percentile (grey dashed lines) also indicate that this interaction is not common in all airstreams that enter the cyclones, but rather occurs in distinct ones.

[Figure]

*(same as above but for the orographically labed cyclones)

As both figures show that the orographic interaction occurs only in some airstreams, we think it is more valuable to split the environmental PV in apveO and apveNO when considering only the orographically influenced cyclones, as we consider it the place to provide the most insights. We will show and mention this in the exemplary trajectory explanation in section 3 and further consider it in the section of orographic PV. Please note, that this is the distribution of all trajectories of orographic cyclones. Therefore, some trajectories acquire negative orographic environmental PV by the time of the mature stage, but the average orographic PV of the corresponding cyclone still provides 25 % of the PV anomaly.

L205 this is the first time you mention the 'core' of the medicane. Is this the same as your 'cyclone effective area' for T=0h? Also do you initialize a back trajectory at every gridpoint in this region? If you do, it isn't clear.

> This was just to use a different term than lower-tropospheric PV anomaly. Trajectories are initialized according to our criteria in Section 2.3, within 200 km around the cyclone center between 975-700 hPa if PV exceeds 0.75 PVU. We are now more explicit in the text to avoid confusion.

Fig 5b could be clearer, I don't think, for example, you explicitly refer to your solid grey PV line (I think it needs to be explicitly said that this is an average of (a) the diagnosed PV over (b) the trajectories).

> Thank you for the remark. We now include an explanation of the grey line in the plain text.

L295: "After the trajectories enter the cyclone effective radius (vertical grey line)". I assume this vertical grey line is the average time the trajectories enter the cyclone effective radius (as they will presumably happen at slightly different times).

> Thank you for the remark. Indeed, the vertical grey line is the time of cyclogenesis and not the average time of trajectories entering the cyclone. We make now an explicit reference of this.

L300: Your 2nd case study involves a cyclone reaching maturity in the Black Sea. I have no problem with this, the tropical-like storms that occasionally form in the Black Sea are structurally similar to medicanes however you don't mention the Black Sea in your introduction. It might be worth including a sentence so the reader is not surprised by the location of this case study.

> The Black Sea is also a hotspot for Mediterranean cyclones and the focus are all Mediterranean cyclones. We think this should not be a surprise for the reader.

L330: The yellow line goes off the scale so I can't see that it produces 2PVU as stated. Also, to be picky, the line looks more yellow than orange. The blue line also goes off the scale. I understand, as a result, we get a zoomed in version of the other lines, but I'm not sure it can justify cropping out this data.

> This is a fair suggestion. We adjusted the scale and changed Fig. 11 accordingly (see comment about diurnal variations after L151 comment. It is 1.85 PVU of convective PV, which we changed in the manuscript accordingly.

L346. A little unclear about the difference between small case apv and APVtot. I'm assuming apv is simply the difference between the diagnosed PV at T+0h and T-48h?

> Yes, indeed apv is simply the difference as indicated by Eq. 4, whereas APVtot refers to Eq. 3. We are now more explicit in the text.